# `ZeroFlow`: Overcoming Catastrophic Forgetting is Easier than You Think

**Tao Feng** [1]  **Wei Li** [1] [*]  **Didi Zhu** [2]  **Hangjie Yuan** [2]  **Wendi Zheng** [1]  **Dan Zhang** [1]  **Jie Tang** [1]

https://zeroflow-bench.github.io/

## Abstract

Backpropagation provides a generalized configuration for overcoming catastrophic forgetting. Optimizers such as SGD and Adam are commonly used for weight updates in continual learning and continual pre-training. However, access to gradient information is not always feasible in practice due to black-box APIs, hardware constraints, or non-differentiable systems, a challenge we refer to as *the gradient bans*. To bridge this gap, we introduce `ZeroFlow`, the first benchmark designed to evaluate gradient-free optimization algorithms for overcoming forgetting. ZeroFlow examines a suite of forward pass-based methods across various algorithms, forgetting scenarios, and datasets. Our results show that forward passes alone can be sufficient to mitigate forgetting. We uncover novel optimization principles that highlight the potential of forward pass-based methods in mitigating forgetting, managing task conflicts, and reducing memory demands. Additionally, we propose new enhancements that further improve forgetting resistance using only forward passes. This work provides essential tools and insights to advance the development of forward-pass-based methods for continual learning.

## 1. Introduction

Catastrophic forgetting remains one of the major challenges on the path to artificial general intelligence (AGI) (Hadsell et al., 2020; Zhou et al., 2023b), *i.e.*, models tend to forget previously learned tasks when trained on new ones on time-evolving data flow (Feng et al., 2022b). This phenomenon is commonly seen across various tasks, including continual learning (CL) (Wang et al., 2023), fine-tuning of foundation models (FMs) (Sun et al., 2025; Yuan et al., 2024), and

---

[*]Core contribution  [1]Tsinghua University [2]Zhejiang University. Correspondence to: Jie Tang <jietang@tsinghua.edu.cn>.

*Proceedings of the 42$^{nd}$ International Conference on Machine Learning*, Vancouver, Canada. PMLR 267, 2025. Copyright 2025 by the author(s).

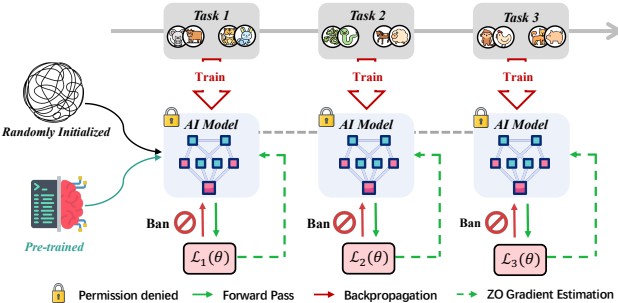

Figure 1: **Illustrations of `ZeroFlow`**. New tasks (or downstream tasks) arrive sequentially, the gradient bans block the model from learning and memorizing using backpropagation. `ZeroFlow` overcome this issue via forward passes.

continual pre-training (CPT) (Shi et al., 2024; Zhu et al., 2024b), etc. Among them, optimization algorithms play a crucial role, *e.g.*, SGD has become the default choice during CL (van de Ven et al., 2022), while Adam is frequently seen in fine-tuning FMs (Luo et al., 2023; Zhu et al., 2024a). These optimization algorithms in tandem with various methods (ranging from regularization and rehearsal strategies to architectural changes) rely on gradient information to avoid forgetting (Zhou et al., 2023c; Bian et al., 2024). Nonetheless, in real-world scenarios, gradient information is not always available or computable (*i.e.*, the gradient bans), like, *Scenario i:* large language models as a service (LLMaaS) and black-box APIs. *Scenario ii:* hardware systems that do not support principled backpropagation. *Scenario iii:* AI for science with non-differentiable underlying systems.

In other words, *Scenario i* implies that pretrained models are monetized (Miura et al., 2024) (model owners do not publicly release their pretrained models but instead the service), *i.e.*, only the inputs and outputs are accessible (Gan et al., 2023; Sun et al., 2022). *Scenarios ii/iii* implies that the limitations prevent or restrict the execution of backpropagation (Lillicrap et al., 2020), *i.e.*, extremely high memory demands (Mangrulkar et al., 2022), unsupported systems and hardware (Jabri & Flower, 1992), or non-differentiable functions, etc (Tavanaei et al., 2019; Gu et al., 2021). The above means that typical methods for overcoming forgetting are not available because backpropagation is banned, as Figure 1. This yields the primary question to be explored,

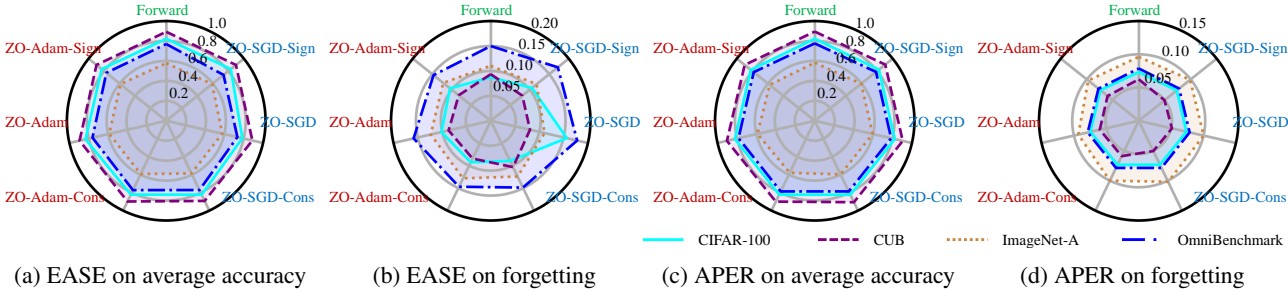

Figure 2: **ZeroFlow Evaluation Results of Catastrophic Forgetting.** We visualize the evaluation results of 2 models (EASE (Zhou et al., 2024b) and APER (Zhou et al., 2023a)) in several **ZeroFlow** dimensions (average accuracy over all tasks and a forgetting metric). For comprehensive numerical results, please refer to Table 1.

---

*(Q) Could we establish a benchmark under gradient bans for overcoming catastrophic forgetting, and explore the overlooked optimization principles?*

To tackle *(Q)*, a natural idea is to use the forward pass-based method (Hinton, 2022; Baydin et al., 2022; Ren et al., 2022) instead of backpropagation to overcome forgetting. The zeroth-order (ZO) optimization methods (Flaxman et al., 2004; Nesterov & Spokoiny, 2017; Malladi et al., 2023; Ghadimi & Lan, 2013), as representative methods, are well-suited to this issue due to their relaxed information requirements, as they rely only on function values rather than gradients. Under gradient bans, DECL and DFCL (Yang et al., 2024) first attempt to overcome forgetting from a stream of APIs, but they focus on synthetic data level rather than optimization. Therefore, it remains elusive whether benchmark studies using gradient-free methods can mitigate forgetting.

In this work, we explore several **Zero**th-order optimization methods on dynamic data **Flow** (as shown in Figure 1), examining their performance across various forgetting scenarios, model types, and evaluation metrics. Through a detailed analysis, we reveal the overlooked potential of forward passes and various ZO methods in overcoming catastrophic forgetting. This benchmark study offers an easier way to overcome forgetting and helps reveal the pros and cons of these methods in alleviating forgetting. Extended from the gained insights, we introduce three new enhancement variants that further improve ZO optimization to overcome catastrophic forgetting. Simply put, we can mitigate forgetting more effectively and efficiently using only forward passes.

**Our rationale** for choosing the ZO optimization algorithms to overcome forgetting for the following two key considerations: (i) implementation cost minimization, that is, we expect minimal modifications to existing optimizers. (ii) theory of diversity, that is, we expect to cover diverse optimization methods. These considerations ensure that our benchmark is comprehensive and simplified. And, an ap-

pealing property is that we need only forward passes to be enough to overcome forgetting. *Maybe, once is all it takes!*

To sum up, our contributions are listed below,

(i) We propose the first benchmark **ZeroFlow** for overcoming forgetting under gradient bans. This benchmark includes our investigations into 7 forward pass optimization algorithms, several forgetting scenarios and datasets with varying complexity, and task sequences (as Figure 2).

(ii) Through this benchmark, we uncover overlooked optimization principles and insights into how forward passes can mitigate forgetting. These include the role of forward passes in managing task conflicts and the trade-offs between forgetting and memory efficiency. We proved that catastrophic forgetting can be overcome in an easier way!

(iii) Apart from a comprehensive evaluation of catastrophic forgetting, we introduce three enhancement techniques, which further improve the performance and efficiency of just forward passes to overcome forgetting.

## 2. Literatures

**Catastrophic forgetting.** Catastrophic forgetting occurs across various tasks, including CL, fine-tuning of FMs, and CPT (Zhou et al., 2023b; Wang et al., 2023; Zhuang et al., 2022a; Luo et al., 2023). To mitigate this issue, various methods have been proposed (Aojun et al., 2025; Jeeveswaran et al., 2023; Sun et al., 2023b; Li et al., 2024). In CL, methods range from regularization and rehearsal strategies to architectural changes (Zhuang et al., 2023; Bian et al., 2024; Lu et al., 2024). Lately, pre-trained models (PTM) further advanced these methods due to their strong generalization (Yuan et al., 2022; Feng et al., 2022a), as seen in PTM-based CL (Zhou et al., 2024a). All these methods share a common goal: achieving an optimal balance between learning plasticity and memory stability (Wang et al., 2023). In FMs, catastrophic forgetting often arises from overfitting to small fine-tuning datasets during CPT or

fine-tuning (Luo et al., 2023; Zhu et al., 2024a). Common techniques to address this include learning rate adjustment, parameter-efficient fine-tuning, mixed data strategies, and instruction tuning (Luo et al., 2023; Zhang et al., 2025). Additionally, as foundational models increasingly gain multi-modal capabilities, the complexity of catastrophic forgetting also intensifies (Zhao et al., 2024a; Zhu et al., 2024a).

**Optimization for catastrophic forgetting.** Two broad categories of optimization methods exist for overcoming forgetting, *(i) Standard Optimization.* SGD and the Adam family are frequently employed to investigate catastrophic forgetting (Hadsell et al., 2020; Masana et al., 2022). For instance, in CL, various CL methods predominantly utilize the SGD optimizer for standard evaluations (van de Ven et al., 2022; Sun et al., 2023a; Zhou et al., 2024c). In fine-tuning the LLM, the Adam series is commonly used to observe forgetting phenomena (Luo et al., 2023; Zhu et al., 2024a). Some works explored orthogonal spaces with these standard optimizers to alleviate forgetting (Lopez-Paz & Ranzato, 2017; Feng et al., 2022c; Saha et al., 2020), such as OGD (Farajtabar et al., 2020), and GPM (Saha et al., 2020). Moreover, other works (Farajtabar et al., 2020; Chaudhry et al., 2018; Lopez-Paz & Ranzato, 2017) modified the gradients in the standard optimization process to align the learning spaces of new and old tasks, such as Uni-Grad (Li et al., 2024). The core of these efforts (Deng et al., 2021; Shi et al., 2021) is to find an equilibrium between learning and forgetting in optimization. *(ii) Sharpness-aware Optimization.* This series of methods (He et al., 2019; Foret et al., 2020; Zhong et al., 2022; Zhuang et al., 2022b) has gained attention due to the effectiveness of the flat minimum in mitigating forgetting (Li et al., 2024; Kong et al., 2023; Cha et al., 2021; Mehta et al., 2023). Methods such as FS-DPGM (Deng et al., 2021), F2M (Shi et al., 2021), DFGP (Yang et al., 2023), SAM-CL (Tung et al., 2023) overcome forgetting in the flatness areas of different configurations. C-Flat (Bian et al., 2024) proposed a CL-friendly general optimization framework, that holds promise as a baseline optimizer for overcoming forgetting.

**Our work.** The works mentioned above are all rooted in a gradient feedback mechanism. Such mechanisms are powerless against catastrophic forgetting without explicit gradient information. Our work overcomes forgetting only via forward pass instead of gradient feedback.

# 3. Exploring Zeroth-Order Optimization to Overcome Forgetting

## C.1. Zeroth-Order Optimization

Zeroth-order (ZO) optimization has been extensively studied over the years within the realms of numerical computation and approximation algorithms. It functions as an alterna-

---

**Algorithm 1** Genetic formulation of ZO optimization

**Require:** Initialized model parameters $\theta_0 \in \Theta \subseteq \mathbb{R}^d$, training dataset $\mathcal{D} = \{(x_i, y_i)\}_{i=1}^m \in \mathcal{X} \times \mathcal{Y}$, empirical loss function $\mathcal{L}$, learning rate $\eta_t$, gradient perturbation vector $\xi$, and descent direction computation $\phi(\cdot)$
1: **while** $\theta_t$ not converged **do**
2:   Sample mini-batch $\mathcal{B}$ from $\mathcal{D}$
3:   **Step 1. ZO gradient estimation:**
4:   $\hat{\mathbf{g}}_t = \hat{\nabla}\mathcal{L}(\theta, \xi; \mathcal{B})$
5:   **Step 2. Descent direction computation:**
6:   $\mathbf{h}_t = \phi\left(\{\hat{\mathbf{g}}_i\}_{i=1}^t\right)$
7:   **Step 3. Parameter updating:**
8:   $\theta_{t+1} = \theta_t - \eta_t \cdot \mathbf{h}_t$
9:   $t = t + 1$
10: **end while**
**Ensure:** Updated model $\theta_t$

---

tive solution for estimating descent directions in scenarios where first-order (FO) gradients are either inaccessible or infeasible to compute. Considering a deep learning model parameterized with $\theta \in \Theta \subseteq \mathbb{R}^d$, and given a mini-batch $\mathcal{B}$ extracted from the training dataset $D = \{(x_i, y_i)\}_{i=1}^m$. Let $L(\theta; \mathcal{B})$ denote the empirical loss, then the genetic formulation of ZO optimization follows Algorithm 1.

**1) ZO gradient estimation.** Randomized Gradient Estimation (RGE (Nesterov & Spokoiny, 2017)) and Coordinate-wise Gradient Estimation (CGE (Berahas et al., 2022)) perturb the model using $\xi$, which is generated either from a random unknown distribution (in RGE) or by modifying individual coordinates (in CGE), and then observe the changes in the loss function $\mathcal{L}$ after each perturbation, step by step, to provide a reliable gradient estimate. However, due to their reliance on slow single-direction perturbation, these methods are not well-suited for deep learning tasks, as performing a full perturbation in high-dimensional parameter spaces is time-consuming. For instance, typical vision models like ResNet trained on ImageNet have over 25 million parameters. Performing per-dimension perturbations over such a large parameter space renders ZO-based querying highly inefficient. Standard Simultaneous Perturbation Stochastic Approximation (SPSA(Spall, 1992)) improves efficiency by generating pairs of symmetric vectors and perturbing in multiple directions simultaneously, as follows,

$$\hat{\nabla}L(\theta, \xi; \mathcal{B}) = \frac{L(\theta + \epsilon\xi; \mathcal{B}) - L(\theta - \epsilon\xi; \mathcal{B})}{2\epsilon}\xi^{-1}. \quad (1)$$

Where $\epsilon$ is a positive scaler and $\xi$ is recommended to follow a symmetric distribution with finite inverse moments (*e.g.*, the Rademacher distribution). The symmetric distribution ensures unbiased exploration of perturbations in both positive and negative directions of parameters at each step. And the finite inverse moments property guarantees that the steps

are well-controlled, avoiding excessively large steps due to $\xi^{-1}$ drawn from the distribution (*e.g.*, $\mathbb{E}[1/|\xi|^p]$ for some large p), which would otherwise lead to an unstable optimization process. In practical implementations for models with a large number of parameters (*e.g.*, MeZO (Malladi et al., 2023) in LLMs (Zhao et al., 2024b)), Gaussian noise with zero mean induces substantial perturbations, thereby enhancing exploration across the parameter space and facilitating the escape from local minima. This methodology achieves gradient estimation with only two objective function evaluations, rendering its computational cost independent of input dimensionality. Such computational efficiency has established SPSA as a preferred method for addressing the complexities of high-dimensional deep learning tasks. While increasing $q$ in $q$-SPSA can improve stability in the update direction, setting $q = 1$ is sufficient for pretrained LLMs (Malladi et al., 2023).

**2) Descent direction computation.** In unconstrained optimization for deep learning, the last gradients $h_t$ generally coincide with the estimated ZO gradients $\hat{g}_t$ (*e.g.*, ZO-SGD (Ghadimi & Lan, 2013), ZO-SCD (Lian et al., 2016)). To reduce approximation errors, ZO-SGD-Sign (Liu et al., 2019) applies an element-wise $sign(\cdot)$ operation. Additionally, ZO-SVRG (Liu et al., 2018), inspired by variance reduction methods in first-order optimization, adjusts the update step by using estimated gradients from previous training examples. CARS (Kim et al., 2021) adaptively selects the smallest function value in each iteration, which helps maintain monotonicity during optimization.

**3) Parameter updating.** Normally, for most ZO methods, parameters are updated in a similar way with FO optimizers, and the learning rate $\eta_t$ is set to constant. Except for the special design for achieving some constraint prerequisites, several methods make an effort to strike a balance between converge speed and accuracy. ZO-AdaMM (Chen et al., 2019) uses an adaptive learning rate and refines gradient estimation by incorporating momentum from past information. This approach is particularly effective in handling complex and evolving optimization landscapes, where the function's behavior may vary over time or be hard to capture with straightforward gradient approximations.

### C.2. Zeroth-Order Optimization for Catastrophic Forgetting

**Rationality.** ZO optimization leverages the function values of the forward passes to approximate FO gradients, making it feasible to avoid gradient bans. This feature enables seamless integration into common forgetting scenarios, such as CL. We explore it in the following three categories.

i) *Memory-based* methods maintain a repository of exemplars from previous tasks and dynamically adjust the overall loss function by combining these stored samples with new

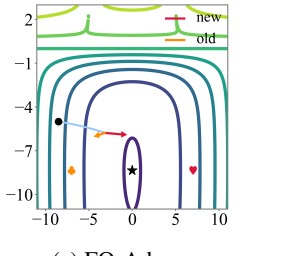
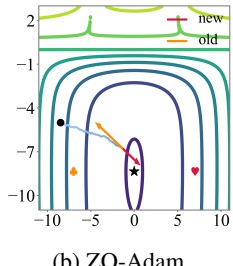

(a) FO-Adam      (b) ZO-Adam

Figure 3: **Trajectory of FO and ZO Optimization during Overcoming Forgetting.** The trajectory is taken when using the total loss from both tasks (cyan) and the gradients from each individual task at fixed points during optimization (red and orange). The trends of ZO optimization hold the potential to manage forgetting and learning.

data based on learning progress.

$$\mathcal{L}_{total} = \frac{1}{N_{context}}\mathcal{L}_{cur} + (1 - \frac{1}{N_{context}})\mathcal{L}_{replay}, \quad (2)$$

where $N_{context}$ represents the number of contexts encountered so far. In Experience Replay (Rolnick et al., 2019), both components use classification loss based on their respective data distributions, so ZO gradients can be expressed as $\hat{\nabla}\mathcal{L}_{cur}$ and $\hat{\nabla}\mathcal{L}_{replay}$ respectively. However, in the emerging generative replay workflows (Shin et al., 2017), Equation (2) may introduce additional loss for the training of generators. In this case, the generator can be trained using standard backpropagation or in conjunction with ZO training without FO gradients.

ii) *Extension-based* methods can be divided into fixed and dynamic architectures. Fixed architectures separate model parameters for specialized context learning, while dynamic architectures expand the model size during adaptation. Both approaches mitigate forgetting from the model's perspective and enable model-agnostic ZO solutions.

iii) *Regularization-based* methods penalize significant changes to parameters important for old tasks or maintain the output distribution with respect to previous inputs. The template loss function is given by

$$\mathcal{L}_{total} = \mathcal{L}_{cur} + \alpha\mathcal{L}_{reg}, \quad (3)$$

where $\alpha$ is a coefficient hyperparameter. The FO gradients from dual objectives ($\mathcal{L}cur$ for adaptation and $\mathcal{L}reg$ for preservation) drive optimization toward their respective optima, achieving inter-task equilibrium. Notably, ZO gradient estimates, though obtained in a noisy environment, exhibit comparable optimization behavior.

As shown in Figure 3, we visualize and compare the optimization trajectories of ZO and FO methods under the learning–memory trade-off dynamics in continual learning. The objective is defined over two-dimensional parameters, with axes specified in Appendix A.2. The striking similarity

Table 1: **ZeroFlow** Evaluation on CIFAR-100, ImageNet-A, CUB and OmniBenchmark. This table compares average accuracy, final accuracy, and forgetting measures of 2 models, and 4 forgetting scenarios. More intuitive trend please see Figure 2. All ZO optimizations use a query budget of $q = 1$. **Bold** indicates the best accuracy achieved among **ZeroFlow**.

| Method | Optimizer | Strategy | CIFAR-100 | | | CUB | | | ImageNet-A | | | OmniBenchmark | | |
|---|---|---|---|---|---|---|---|---|---|---|---|---|---|---|
| | | | Avg | Last | Fgt | Avg | Last | Fgt | Avg | Last | Fgt | Avg | Last | Fgt |
| EASE | SGD | FO | 91.23 | 85.96 | 7.32 | 89.31 | 83.76 | 9.61 | 61.24 | 51.02 | 10.84 | 74.73 | 67.40 | 15.11 |
| | | ZO | 78.62 | 68.40 | 15.64 | 88.94 | 82.91 | 8.08 | 57.87 | 48.32 | 11.08 | 73.50 | 66.60 | 17.78 |
| | | Sign | **83.21** | **75.88** | 10.58 | **89.81** | **84.61** | 8.10 | **59.15** | **49.31** | 11.77 | 73.81 | 66.75 | 17.21 |
| | | Conserve | 82.22 | **75.88** | 8.93 | 89.21 | 83.42 | 10.31 | 58.61 | 48.58 | 12.41 | **77.07** | **70.73** | 14.87 |
| | Adam | FO | 90.56 | 84.82 | 7.69 | 84.44 | 77.10 | 10.51 | 59.60 | 47.20 | 19.08 | 74.27 | 66.28 | 15.63 |
| | | ZO | **83.36** | **76.09** | 10.16 | 89.49 | 84.14 | 8.67 | 58.90 | 48.72 | 12.35 | 76.15 | 69.69 | 15.87 |
| | | Sign | 83.14 | 76.01 | 10.44 | **89.82** | **84.65** | 8.21 | 58.97 | **48.85** | 12.20 | 77.12 | **71.08** | 14.68 |
| | | Conserve | 82.15 | 75.65 | 9.24 | **89.82** | 84.61 | 8.40 | **59.23** | **48.85** | 12.81 | **77.19** | 70.99 | 14.68 |
| | - | Forward | 82.26 | 76.05 | 8.74 | 89.26 | 83.67 | 9.35 | 57.76 | 48.19 | 11.03 | 77.00 | 70.74 | 14.99 |
| APER | SGD | FO | 82.31 | 76.21 | 7.33 | 90.56 | 85.16 | 5.19 | 59.50 | 49.37 | 9.91 | 78.61 | 72.21 | 7.87 |
| | | ZO | **82.33** | 76.21 | 7.36 | 90.53 | 85.20 | 5.12 | 59.58 | 49.51 | 10.02 | 78.60 | 72.21 | 7.85 |
| | | Sign | 82.32 | **76.23** | 7.32 | 90.42 | **85.28** | 4.96 | 59.65 | **49.77** | 9.89 | 78.60 | **72.26** | 7.78 |
| | | Conserve | 82.31 | 76.21 | 7.33 | **90.62** | **85.28** | 5.05 | **59.68** | 49.70 | 10.18 | **78.61** | 72.21 | 7.87 |
| | Adam | FO | 82.31 | 76.21 | 7.33 | 90.56 | 85.16 | 5.19 | 59.60 | 49.77 | 10.06 | 76.60 | 72.21 | 7.85 |
| | | ZO | 82.12 | 75.45 | 7.47 | **90.33** | 84.31 | 6.01 | **58.89** | **49.24** | 9.32 | 78.44 | 72.10 | 7.87 |
| | | Sign | 82.01 | 75.60 | 7.38 | 89.86 | 84.18 | 5.99 | 57.82 | 48.12 | 9.72 | 78.26 | 72.05 | 7.75 |
| | | Conserve | 82.21 | 75.98 | 7.34 | 89.96 | **84.48** | 5.90 | 57.86 | 47.53 | 10.00 | **78.61** | 72.21 | 7.87 |
| | - | Forward | **82.32** | **76.22** | 7.32 | 89.47 | 83.38 | 6.24 | 58.25 | 47.99 | 9.62 | 77.61 | 71.45 | 7.87 |

between the two trajectories highlights the potential of ZO optimization in effectively balancing learning and forgetting, thereby motivating our further investigation.

**Potential.** The intrinsic optimization mechanism of ZO exhibits particular promise in continual learning scenarios. Intuitively, ZO perturbs parameters using random or coordinate-wise directional vectors and observes changes in the evaluation function, effectively optimizing within a noisy environment. This approach enables small parameter modifications to yield significant impacts on target objectives, resulting in distinctive gradient estimations compared to FO optimization. Notably, while ZO methods do not explicitly incorporate sharpness regularization terms, they naturally facilitate the exploration of flat regions in parameter space. The influence of optimizing flat regions with ZO approaches in continual learning can be summarized in two main manifolds: (i) For previous tasks, the noise-induced parameter robustness enhances resilience against perturbations from new task adaptation; (ii) For new tasks, empirical evidence suggests that convergence to flat minima generally leads to lower generalization error.

**Risk.** Although ZO demonstrates superior generalization abilities, its practical performance is limited by optimization strategies and the complexity of the optimization setting. Despite significant efforts to reduce convergence error, optimizing models from scratch in high-dimensional space remains challenging due to slow convergence speed (proportional to the parameter dimension $d$). For instance, origin CGE-based ZO training for a model with 12k parameters takes 70.32 hours in DeepZero (Chen et al., 2023). Such computational demands render from scratch training impractical for high-dimensional CL models, particularly those employing expansion-based architectures. Consequently, we focus our discussion on leveraging ZO optimization to overcome forgetting within a pre-training context.

## 4. **ZeroFlow** Benchmark

This section delves into the empirical performance of ZO optimization in overcoming catastrophic forgetting. Our ZeroFlow benchmark evaluates average performance across incremental stages, final-stage accuracy, forgetting, and efficiency, while accounting for dataset complexity and model diversity.

### D.1. Benchmark Setups

**Forgetting scenarios, schemes, and models.** We conduct evaluations under a standard catastrophic forgetting setting, namely class incremental learning. For this purpose, we investigate two state-of-the-art schemes: EASE and APER. Both models are initialized with ViT-B/16 pretrained on ImageNet-1K (IN1K), and are subsequently fine-tuned on four downstream tasks of varying complexity—ranging from standard benchmarks such as CIFAR-100 and CUB, to more challenging datasets like ImageNet-A and OmniBenchmark, which exhibit a large domain gap from the pretraining distribution (Zhou et al., 2024a;c). Following (Zhou et al., 2023a), each dataset is evenly split into 10

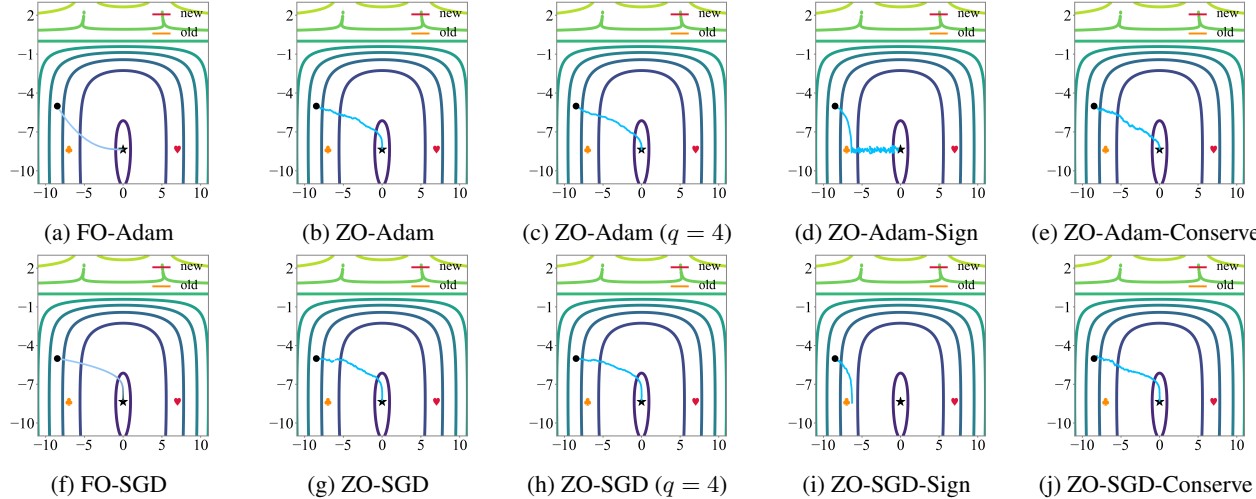

Figure 4: **The Trajectory of Different Optimization during Overcoming Forgetting.** ♥, ♠, and ★ denote the minima for the new, old, and both tasks, respectively. The trajectory is taken when using the total loss from both tasks (cyan).

incremental tasks by class. For instance, OmniBenchmark contains 300 classes, with 30 classes introduced at each stage. No memory is permitted for storing past examples.

**Benchmark setup and details.** To evaluate the application of `ZeroFlow` in forgetting scenarios, we include the methods described in Section C.1, specifically ZO (Ghadimi & Lan, 2013), Sign (Liu et al., 2019), and Conserve (Kim et al., 2021; Zhang et al., 2024), in comparison with their FO counterparts using SGD and Adam optimizers (Chen et al., 2019). Additionally, as highlighted in (Zhang et al., 2024), Forward-Grad (Baydin et al., 2022) which relies on forward mode automatic differentiation, potentially becomes a missing but competitive forward pass baseline. In a nutshell, `ZeroFlow` covers 7 forward pass-based methods: ZO-SGD, ZO-SGD-Sign, ZO-SGD-Conserve, ZO-Adam, ZO-Adam-Sign, ZO-Adam-Conserve, Forward-Grad. Unless otherwise specified, the query budget is fixed to 1 for efficiency. Notably, here we consider generating one set of perturbation vectors for the entire model as one query. In other words, we usually require 2 forward propagations for two-point finite difference gradient estimations.

**Evaluation metrics.** Overall, we adopt two categories of evaluation metrics in `ZeroFlow`: accuracy and efficiency. The accuracy metrics include average accuracy across all tasks, final-task accuracy, and a forgetting score (BWT in Appendix B.5). The efficiency metrics encompass memory usage (GPU), query budget, and runtime. Together, these metrics provide insights into the resource demands of ZO optimization for mitigating forgetting.

### D.2. Evaluation Results of `ZeroFlow`

**`ZeroFlow` evaluation on continual learning.** In Table 1, we evaluate the performance of different BP-free and BP-

based (FO-SGD and FO-Adam) methods in a typical forgetting scenario (continual learning). We use two SOTA models as examples (EASE (Zhou et al., 2024b) and APER (Zhou et al., 2023a)) and investigate SGD and Adam optimizers, 7 forward pass-based methods, and four commonly used datasets. Several observations are listed below,

First, the performance of ZO method is comparable to or even surpasses that of the FO method across almost all forgetting metrics and datasets. However, as will be shown later, the FO method requires significantly more memory overhead. This suggests that forward passes alone can effectively mitigate forgetting, and the ZO method offers a simpler, more efficient alternative. In some cases, such as with ZO-Adam and ZO-SGD on OmniBenchmark, ZO methods even outperform FO methods.

Second, Forward Grad demonstrates competitive performance when compared to other ZO and FO methods. Unlike typical ZO methods, Forward Grad utilizes a unique forward pass mechanism, making it a promising baseline for future studies. A more intuitive trend in overcoming forgetting refer to Figure 6. These observations motivate further exploration into the effectiveness of ZO method.

**`ZeroFlow` helps manage memory and runtime.** In Table 2, we compare the efficiency of various ZO and FO optimizers in mitigating catastrophic forgetting, focusing on two key aspects: memory cost (in GB) and runtime cost (in seconds). First, naive ZO optimization reduces memory usage by approximately fivefold compared to FO optimization. Moreover, ZO methods reduce runtime per iteration by around 50% relative to FO, significantly improving their practicality for overcoming forgetting. Notably, we regenerate the perturbation vectors for model parameters iteratively by storing random seeds. This degrades the vec-

Table 2: **Memory Cost (GB) and Runtime Cost (s) of Each Optimizer on 3 Forgetting Scenarios.** The per-epoch runtime in seconds (s). ZO-SGD w/ query budget $q = 1, 4$ and all other optimizers w/ query budget $q = 1$.

| Optimizer | Memory ⇓ | CIFAR-100 | CUB | ImageNet-A |
|---|---|---|---|---|
| FO-SGD | 12.08 GB | 59.3s | 16.1s | 12.2s |
| ZO-SGD ($q = 1$) | 2.41 GB | 32.4s | 8.3s | 6.8s |
| ZO-SGD ($q = 4$) | 2.41 GB | 111.7s | 28.7s | 18.0s |
| ZO-SGD-Sign | 2.41 GB | 32.4s | 8.3s | 6.8s |
| ZO-SGD-Conserve | 2.41 GB | 70.1s | 15.7s | 12.4s |
| Forward-Grad | 3.94 GB | 45.9s | 11.1s | 9.0s |

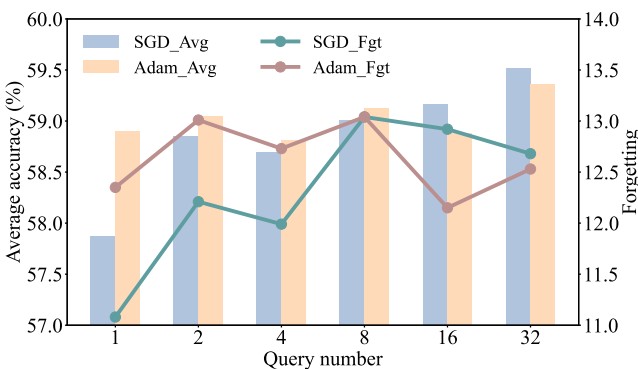

Figure 5: **Performance Comparison under Different Query Numbers**. Both optimizers show improved performance as query numbers increase.

tor granularity from full-model to per-layer level, thereby further reducing the memory required for forward evaluations in **ZeroFlow**, at the cost of additional runtime for regenerating the vectors. Second, the ZO and Sign variants demonstrate comparable efficiency in both memory and runtime. Although increasing the number of queries can impact runtime efficiency, it does not compromise memory advantages. Third, Conserve also demonstrates efficient memory management, although its runtime is approximately twice as long as that of naive ZO. This may partly explain its stronger performance in some scenarios, as shown in Table 1. Finally, the Forward Gradient method requires more memory than other ZO-based approaches because it involves computing gradients via the Jacobian-vector product (JVP), which necessitates storing all intermediate activations during the forward pass. For models like ViT, this includes large attention maps and other intermediate representations. In contrast, naive ZO methods only require two forward passes on perturbed inputs and avoid storing these intermediate values, resulting in much lower memory usage.

**Trade-off between performance and query number.** As shown in Figure 5, we investigate the impact of query numbers on optimization performance, comparing SGD and Adam optimizers in the zeroth-order setting. Both optimizers demonstrate improved performance as query numbers

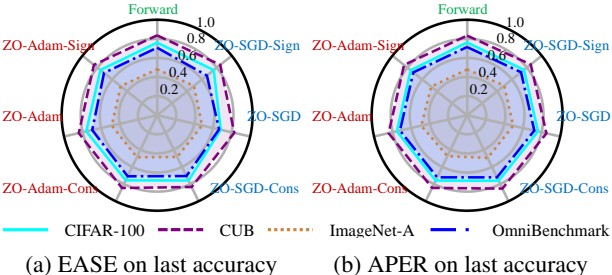

| (a) EASE on last accuracy | (b) APER on last accuracy |
|---|---|

Figure 6: **ZeroFlow Evaluation Results for Forgetting.** We visualize the evaluation of 2 models in last-task accuracy.

increase across $\{1,2,4,8,16,32\}$, suggesting that additional function evaluations enable more accurate gradient estimation. The results suggest that in scenarios where function evaluation costs are manageable, higher query numbers can yield substantially better performance, with Adam being particularly effective at leveraging the additional gradient information for enhanced optimization outcomes.

## 5. Insights and Discussions

As shown in Figure 4, we visualized the optimization trajectories of both forward passes and backpropagation methods. Our analysis reveals several key insights:

**Convergence behavior across optimizer families.** In Figure 4, both FO and ZO methods demonstrate successful convergence to the minima of new and old knowledge spaces, regardless of whether they use Adam or SGD as their base optimizer. This convergence consistency validates our theoretical foundation.

**Distinct trajectory characteristics of FO and ZO.** FO approaches (Figure 4a, 4f) show smoother optimization paths due to their access to exact gradient information. In contrast, ZO methods demonstrate varying degrees of exploration behavior through trajectory jitter. This exploration pattern is particularly pronounced in ZO-Adam variants compared to ZO-SGD variants, indicating that the base optimizer choice significantly influences the exploration-exploitation trade-off during optimization.

**Path characteristics in ZO optimization.** Comparing base ZO methods with their $q = 4$ counterparts (Figure 4b vs 4c, Figure 4g vs 4h), we observe that increasing query numbers leads to smoother trajectories, suggesting that more queries help provide more stable gradient estimates. The Sign variants (Figure 4d, 4i) demonstrate more pronounced oscillations in their trajectories, particularly visible in the ZO-Adam-Sign case. In contrast, the conservative variants (Figure 4e, 4j) maintain relatively stable paths that better balance between the old and new task minima.

**Distinct characteristics between optimizer families.** Adam-based approaches (Figure 4a–4e) demonstrate more

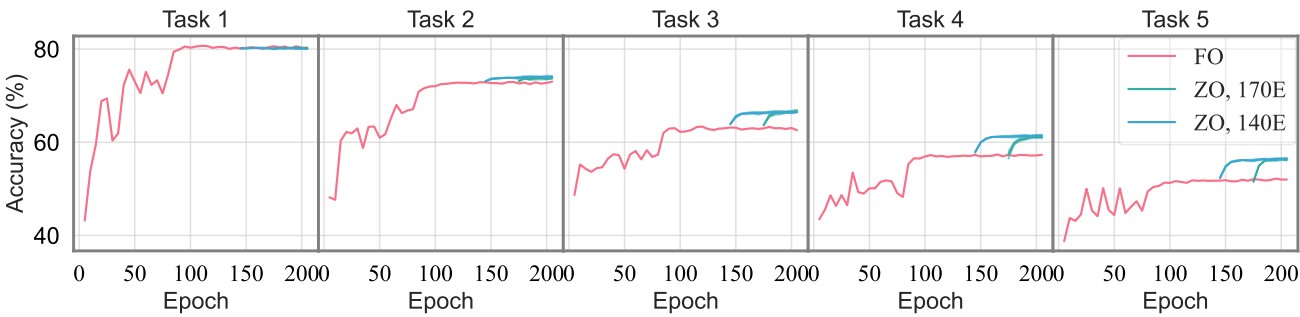

Figure 7: **Effectiveness of Hybrid ZO in Overcoming Forgetting.** In Hybrid ZO, backward benefits from forward passes.

Table 3: **Effectiveness of Historical Estimation in Mitigating Forgetting.** Proportion of 0% denotes that the plain optimizer ZO-SGD. **Bold** indicates the best performance.

| Metrics | Proportion | | | | |
|---------|------|-------|-------|-------|-------|
| | 0% | 20% | 40% | 60% | 80% |
| Avg | 57.87 | **58.90** | 58.76 | 58.34 | 57.83 |
| Last | 48.32 | **49.04** | 48.84 | 48.42 | 48.10 |
| Fgt | 11.08 | 11.79 | 11.78 | 11.60 | 11.57 |

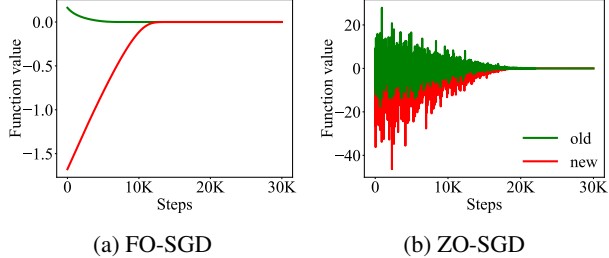

| (a) FO-SGD | (b) ZO-SGD |
|---|---|

Figure 8: **Variation in Function Values of Forward Passes.** Function values for new tasks is highlighted in red, old tasks is highlighted in green.

oscillatory trajectories with frequent direction adjustments, indicating a more dynamic exploration of the loss landscape. In contrast, SGD-based methods (Figure 4f–4j) exhibit smoother and more stable trajectories, suggesting a more gradual progression toward the optimization objective. These distinct optimization patterns could influence how each method balances between preserving old task knowledge and adapting to new tasks.

# 6. New Enhancement to Mitigate Forgetting

In ZO optimization, the estimation of the gradients relies on a finite difference of the objective function. We set query budget $q = 1$ in the benchmark for efficiency. However, limited queries cannot capture the accurate ZO directions. When the model learns tasks sequentially, the high variance inherent in ZO gradient estimation poses a critical challenge. Though increasing query numbers can stabilize the gradient estimates, it leads to prohibitive overhead *Thus, exploring variance-reduced optimization algorithms is crucial for ZO-based CL.* Specifically, we propose 3 enhancements to stabilize the ZO optimization process:

**Enhancement 1: Hybrid ZO to overcome forgetting.** While ZO methods does not explicitly minimize sharpness, it stabilizes optimization by approximating gradients and assessing the rate of change in loss function through perturbations. This indirect approach helps reduce the curvature of the loss landscape, steering the optimization away from sharp and unstable regions. This insight motivates us to investigate Hybrid ZO method. Figure 7 illustrates results

hybrid ZO. We first use FO to coarsely optimize to a local minimum (first 140 or 160 epochs) and then refine the solution by searching for flatter regions around it using ZO (last 30 or 60 epochs). As the first two subfigures in Figure 7, ZO provides only limited gains to FO. This is because FO inherits strong generalization from the pretrained backbone but loses its generalization ability quickly after two incremental stages. In later stages, ZO helps to remedy the vulnerabilities of backbone trained by FO, leading to significant enhancements compared to the FO baseline.

**Enhancement 2: Leverage historical information to overcome forgetting.** When learning new tasks, models leverage previously learned parameters while prioritizing the preservation of crucial parameters for old tasks. To mitigate interference from new tasks, we propose reweighting old task gradients with historical gradients, which can stabilize perturbations caused by low query loops in ZO optimization. Figure 8 illustrates the function value trajectories for both old and new tasks. While FO optimization shows smooth convergence toward the global optimum, ZO optimization exhibits a more volatile path. Notably, objectives related to old tasks demonstrate smaller changes in both magnitude and variance. This observation motivates us to stabilize the optimization by reducing changes to old gradients through a linear combination with historical gradients: $g_{old} = (1 - \alpha)g_{old} + \alpha g_{historical}$, where larger $\alpha$ indicates greater reliance on historical information for stability, at the cost of reduced contrast with new task gradients.

Table 4: **Effectiveness of Sparsity-induced Estimation in Overcoming Forgetting.** Proportion of 0% denotes the plain ZO-SGD. **Bold** indicates the best performance.

| Ratio | 0% | 10% | 20% | 30% | 40% | 50% | 60% | 70% | 80% | 90% |
|-------|------|------|------|------|------|------|------|------|------|--------|
| Avg | 57.87 | 59.17 | 59.46 | 59.29 | 59.39 | 59.45 | 59.26 | 59.39 | 59.38 | **59.47** |
| Last | 48.32 | 48.58 | 49.05 | 48.72 | 48.91 | **49.24** | 49.11 | 49.05 | 49.11 | **49.24** |
| Fgt | 11.08 | 12.65 | 12.17 | 12.76 | 12.53 | 12.37 | 12.36 | 12.54 | 12.46 | 12.33 |

Table 5: **Ablation Studies on the Effectiveness of Combining Enhancements.**

| Optimizer | Hybrid | Historical | Sparsity | Avg | Last |
|-----------|--------|------------|----------|-----------|-----------|
| FO-SGD | - | - | - | 61.24 | 51.02 |
| ZO-SGD | - | - | - | 57.87 | 48.32 |
|  | ✓ |  |  | 61.40(+3.53) | 51.34(+3.02) |
|  |  | ✓ |  | 58.90(+1.03) | 49.04(+0.72) |
|  |  |  | ✓ | 59.47(+1.60) | 49.24(+0.92) |
|  | ✓ | ✓ | ✓ | 62.07(+4.20) | 51.94(+3.62) |

In Table 3, we validate the effectiveness of historical estimation in mitigating catastrophic forgetting. Modest proportions of historical information (*e.g.*, 20%, 40%, 60%)) outperform ZO-SGD (0%), effectively controlling perturbations while maintaining a low query budget ($q = 1$).

**Enhancement 3: Sparsity-induced estimation helps to overcome forgetting.** In ZO optimization, the gradients for new tasks are often highly uncertain due to the approximation nature of the gradient estimation. To reduce this variance, we implement random sparsification by creating a seed-based mask and setting gradients outside the mask to zero. By reducing the number of non-zero gradient components, we aim to stabilize the optimization process and mitigate the noise in gradient updates.

In Table 4, we report the performance of sparsity-induced ZO in overcoming forgetting. The sparsity level is varied in this experiments, ranging from 10% to 90%. We observe that the sparse technique improves the average and last accuracy across all scales, which implies that forgetting is effectively controlled. The reduction in volatility can be attributed to the sparse strategy yielding smoother gradient estimates compared to plain ZO-SGD, effectively bounding variance to a low level and thus mitigating forgetting. Moreover, the robust performance across different sparsity ratios provides strong evidence for the efficacy of variance control in addressing forgetting.

**Complementary Enhancements:** The results in Table 5 demonstrate that the proposed enhancements are not mutually exclusive and can be effectively integrated. Specifically, FO training can substantially benefit from subsequent fine-tuning with hybrid ZO optimization, as illustrated in Figure 7. Notably, the inherent instability of ZO with large step

fluctuations can sometimes facilitate escaping local minima and encourage broader exploration, which in turn benefits FO convergence. Furthermore, incorporating historical gradients and sparsity perturbations contributes to mitigating forgetting and stabilizing the optimization process.

# 7. Conclusion

This paper introduces **`ZeroFlow`**, a benchmark study that probes a series of forward pass-based methods for overcoming catastrophic forgetting. This work resorts to an easier way (no need for backpropagation and activation storage) to overcome forgetting. Concretely, our benchmarks include various forward pass-based methods, forgetting scenarios, and evaluation metrics. We also reveal the overlooked optimization principles for overcoming forgetting via forward passes. Based on these insights, we propose two easier and better enhancement to overcome forgetting and extend the application of related methods easily.

# Impact Statement

This paper presents work whose goal is to advance the field of Machine Learning. There are many potential societal consequences of our work, none which we feel must be specifically highlighted here.

# Acknowledgments

This work was supported in part by the National Natural Science Foundation of China (NSFC) under Grant 62495063. This work was supported in part by the China Postdoctoral Science Foundation under Grant 2024M761677.

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

# ZeroFlow: Overcoming Catastrophic Forgetting is Easier than You Think
## Supplementary Material

## A. Experimental Details

In this section, we provide an overview of zeroth-order optimization algorithms and the function settings used for the trajectory analysis.

### A.1. Concise Overview of Zeroth-Order Estimation

Zeroth-order optimization aims to minimize/maximize an objective function $f : \mathbb{R}^n \to \mathbb{R}$ without derivative information. The core problem is formulated as $\min_{\theta \in \mathbb{R}^n} L(\theta)$, where $\theta$ denotes the optimization variable. To enable gradient-based updates, Simultaneous Perturbation Stochastic Approximation (SPSA(Spall, 1992)) is a commonly used technique to approximate gradients by perturbing the input variables. Specifically, the gradient $\hat{\nabla} L(\theta)$ at point $\theta$ is estimated as:

$$\hat{\nabla} L(\theta, \xi; B) = \frac{L(\theta + \epsilon\xi; B) - L(\theta - \epsilon\xi; B)}{2\epsilon} \cdot \xi^{-1}, \tag{4}$$

where $\xi \sim \mathcal{N}(\mathbf{0}, \boldsymbol{I})$ is a random perturbation vector, and $\epsilon > 0$ is a small perturbation step size (typically adjusted during optimization).

**ZO-SGD(Ghadimi & Lan, 2013):** Using the gradient estimator $\hat{\nabla} L(\theta, \xi; B)$, zeroth-order algorithms, such as ZO-SGD, follow the iterative update rule:

$$\theta_{t+1} = \theta_t - \eta_t \cdot \hat{\nabla} L(\theta_t, \xi_t; B), \tag{5}$$

where $\eta_t$ is the learning rate at step $t$. ZO-SGD bypasses explicit gradient computation through local function evaluations, making it suitable for high-dimensional, non-convex optimization problems.

**ZO-SGD-Sign(Liu et al., 2019):** A variant of ZO-SGD, known as ZO-SGD-Sign, improves upon the original approach by approximating the gradient direction using the sign of the gradient estimate. The update rule becomes:

$$\theta_{t+1} = \theta_t - \eta_t \cdot \text{sign}(\hat{\nabla} L(\theta_t, \xi_t; B)), \tag{6}$$

where $\text{sign}(\cdot)$ denotes the element-wise sign function. This approach often leads to faster convergence in some problems where the magnitude of the gradient is not as important as its direction.

**ZO-SGD-Conserve(Bergou et al., 2020):** ZO-SGD-Conserve is another variant that conservatively selects the update direction by locally comparing three candidate points, rather than directly committing to a single gradient step. The update rule for this method is:

$$\theta_{t+1} = \arg \min_{y \in \mathcal{C}_t} f(y), \quad \mathcal{C}_t = \left\{ \theta_t, \ \theta_t - \eta_t \cdot \hat{\nabla} L(\theta_t, \xi_t; B), \ \theta_t + \eta_t \cdot \hat{\nabla} L(\theta_t, \xi_t; B) \right\}, \tag{7}$$

This method mitigates overly aggressive updates by evaluating possible directions and choosing the one that locally minimizes the objective function.

**ZO-Adam(Zhang et al., 2024):** ZO-AdaMM (Chen et al., 2019) is the first attempt to apply the Adam family (specifically AMSGrad(Reddi et al., 2019)) to zeroth-order (ZO) optimization algorithms, providing convergence guarantees for both convex and nonconvex settings. The update rule is given by:

$$\theta_{t+1} = \theta_t - \eta_t \cdot \frac{m_t}{\sqrt{V_t} + \epsilon}, \quad V_t = \text{Diag}(\max(v_t, v_{t-1})), \tag{8}$$

$$m_t = \beta_1 m_{t-1} + (1 - \beta_1)\hat{\nabla} L(\theta_t, \xi_t; B), \quad v_t = \beta_2 v_{t-1} + (1 - \beta_2)(\hat{\nabla} L(\theta_t, \xi_t; B))^2,$$

*In our implementation, we simply replace SGD with Adam for convenience, referring to this variant as ZO-Adam. Neverthe-less, we also provide a reference implementation of the original oracle ZO-AdaMM algorithm.*

**Forward Gradient Descent (FGD)(Baydin et al., 2022):** FGD replaces backpropagation with forward-mode automatic differentiation to estimate gradient directions using Jacobian-vector products (JVPs). Instead of computing full gradients via reverse-mode automatic differentiation (AD), FGD samples probe vectors to construct unbiased estimators of the gradient direction. A typical FGD update step is:

$$\theta_{t+1} = \theta_t - \eta_t \cdot \text{JVP}_{\theta_t}(v_t) = \theta_t - \eta_t \cdot \left. \frac{df(\theta)}{d\theta} \cdot v_t \right|_{\theta=\theta_t}, \tag{9}$$

where $v_t$ is a random probe vector (e.g., Rademacher or Gaussian), and $\text{JVP}_{\theta_t}(v_t)$ represents the forward-mode gradient approximation in direction $v_t$. FGD enables training when reverse-mode AD is impractical or unavailable, and offers flexibility for hardware or software systems that only support forward execution. *We denote Forward as FGD throughout this paper.*

### A.2. Function Settings

Following the setup in CAGrad (Liu et al., 2021), we visualize the optimization behavior of first-order (FO) and zeroth-order (ZO) methods in mitigating forgetting. Specifically, we consider a two-dimensional parameter space $\theta = (\theta_1, \theta_2) \in \mathbb{R}^2$, with the following task-specific loss functions: $L_1(\theta) = c_1(\theta)f_1(\theta) + c_2(\theta)g_1(\theta)$ for the old task (orange), and $L_2(\theta) = c_1(\theta)f_2(\theta) + c_2(\theta)g_2(\theta)$ for the new task (red). The parameter point is initialized at $[-8.5, -5]$ to be closer to old tasks, facilitating better adaptation to them. The contour plot in Figure 3 illustrates the overall objective function defined as $L(\theta) = L_1(\theta) + L_2(\theta)$, where the $x$- and $y$-axes correspond to $\theta_1$ and $\theta_2$, respectively.

$$f_1(\theta) = \log\left(\max\left(|0.5(-\theta_1 - 7) - \tanh(-\theta_2)|, \ 5 \times 10^{-6}\right)\right) + 6,$$

$$f_2(\theta) = \log\left(\max\left(|0.5(-\theta_1 + 3) - \tanh(-\theta_2 + 2)|, \ 5 \times 10^{-6}\right)\right) + 6,$$

$$g_1(\theta) = \frac{(-\theta_1 + 7)^2 + 0.1(\theta_2 - 8)^2}{10} - 20,$$

$$g_2(\theta) = \frac{(-\theta_1 - 7)^2 + 0.1(\theta_2 - 8)^2}{10} - 20,$$

$$c_1(\theta) = \max\left(\tanh(0.5 \cdot \theta_2), \ 0\right), \quad c_2(\theta) = \max\left(\tanh(-0.5 \cdot \theta_2), \ 0\right).$$

## B. Additional Results

### B.1. Comprehensive Analysis of Memory Usage on `ZeroFlow`

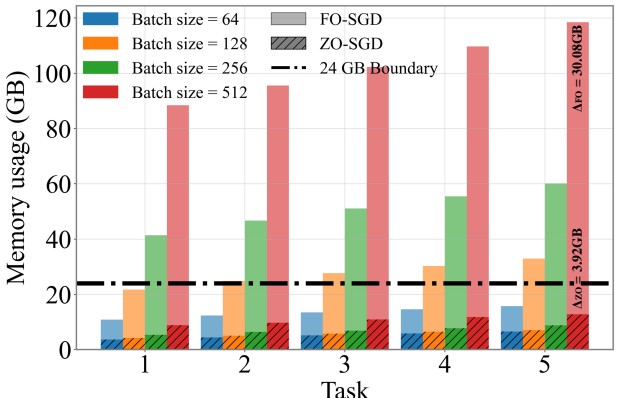

Figure 9: **Comparison of Memory Usage between FO-SGD and ZO-SGD with Different Batch Sizes.** $\Delta$ denotes the increase in memory usage of the final task compared to the initial one.

In this subsection, we provide a detailed comparison of memory usage during incremental learning to demonstrate the storage efficiency of **`ZeroFlow`** (ZO-SGD) compared to its counterpart, FO-SGD. Figure 9 illustrates the peak memory usage of MEMO when trained on the CIFAR-100 dataset. The backbone is fixed as a pretrained ViT-B/16-IN1K model, which is subsequently fine-tuned with batch sizes ranging from 64 to 512. The experimental results highlight the following observations:

**First**, doubling the training batch size significantly increases the memory consumption of FO-SGD, requiring more GPU resources. For instance, completing the entire incremental training process on FO requires 1, 2, 3, and 6 GPUs, respectively, for batch sizes of 64, 128, 256, and 512, with each GPU equipping with 24GB of memory. In contrast, ZO-SGD training consistently requires only one GPU resource.

**Second**, as training progresses, the memory demands for larger batch sizes increase rapidly. For FO, the memory consumption for 512 batches at stage 5 grows by 30.08 GB compared to the initial stage. In contrast, ZO-SGD shows a modest increase of only 3.92 GB, maintaining a low growth rate. As training advances, the memory efficiency of ZO-SGD becomes more pronounced, especially for model-expansion based CL models.

## B.2. More Observations on Optimization Trajectories during Overcoming Forgetting

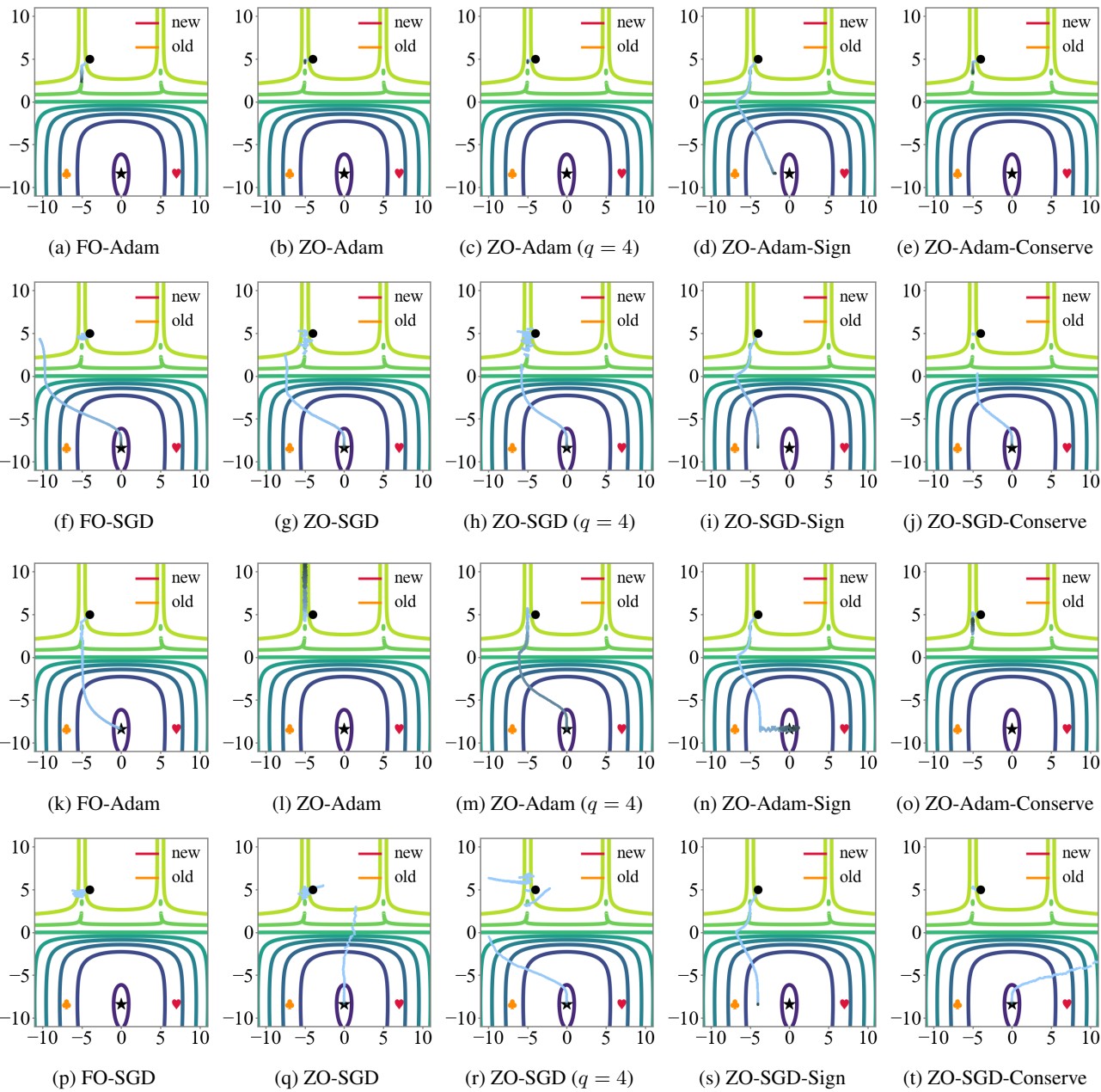

Figure 10: **The Trajectory of Different Optimization during Overcoming Forgetting.** The first and last two rows are trained for 100k steps with learning rates of 0.001 and 0.01, respectively. Red denotes the minimum of new task, orange denotes the minimum of old task. The cyan trajectory taken when using the total loss from both tasks.

In this subsection, we present a different scenario where the model is initialized at a local minimum $\theta_1, \theta_2 = \{-4.0, 5.0\}$, surrounded by intricate valleys, but training with different learning rate as shown in Figure 10. For a learning rate of

0.001, the first-row subfigures demonstrate that Adam using both FO and `ZeroFlow` stagnate in the valley. Even with bias correction, the Adam optimizer still fails to escape the local region without sufficient momentum. However, ZO-Adam-Sign seems to successfully optimize towards the region around the global minimum. Unlike ZO-Adam, ZO-Adam-Sign applies the gradient using a sign function, which outputs either +1 or -1 depending on the gradient direction. This discrete update method, which lacks continuous gradient information, causes ZO-Adam-Sign to take larger, step-like jumps. Particularly in the early stages, where gradient information is sparse or noisy, this leads to more fluctuations and introduces greater randomness in the optimization process, helping it to cross over the valleys. The second-row subfigures use SGD as the base optimizer. We observe that, except for ZO-SGD-Sign, both `ZeroFlow` and FO-SGD converge effectively. This can be attributed to SGD's simple update rule based on function values. Notably, FO-SGD escapes the valley by leaping to a higher and flatter region, while `ZeroFlow` demonstrates the ability to traverse beneath valleys. With a higher learning rate of 0.01, FO-Adam, ZO-Adam with four queries, and ZO-Adam-Sign escape the local region more easily. However, ZO-Adam still stagnates along the valley, demonstrating the stabilizing effect of multiple query loops. Similarly, ZO-Adam-Conserve suffers from the risk of an overly conservative strategy. ZO-SGD also fails to converge to the optimum due to gradient explosion caused by the large learning rate. In contrast, `ZeroFlow` shows minimal degradation despite its inherent randomness.

As a result, the behavior of `ZeroFlow`—sometimes escaping the valley but failing to converge to the optimum, and sometimes getting trapped with low query counts but not with higher ones—highlights the trade-off between randomness and stability during updates. With larger search loops, lower learning rates, and more stable update steps, the model becomes increasingly prone to getting stuck in local minima, especially in continual learning scenarios where balancing old and new tasks introduces additional complexity.

### B.3. Extra Evaluation on Memory Replay Methods

We further evaluate the performance of `ZeroFlow` when applied to a representative replay-based method (MEMO (Zhou et al., 2023c), replay buffer = 2000), to demonstrate its broader applicability. As shown below, `ZeroFlow` consistently remains stable in mitigating forgetting. Notably, although the average accuracies exhibit slight gaps compared to FO methods, the accuracies at the final stage progressively approach or even surpass those of the FO baselines on the CIFAR-100 dataset.

Table 6: **Accuracy Results on MEMO.**

| Method | Optimizer | Strategy | CIFAR-100 | | ImageNet-A | |
|---|---|---|---|---|---|---|
| | | | Avg | Last | Avg | Last |
| MEMO | SGD | FO | 87.43 | 79.66 | 53.15 | 38.97 |
| | | ZO | **85.92** | 79.00 | 46.87 | 25.81 |
| | | Sign | 85.72 | 79.10 | **53.31** | **38.18** |
| | | Conserve | 85.86 | **79.20** | 47.20 | 28.51 |
| | Adam | FO | 86.45 | 76.17 | 54.06 | 41.54 |
| | | ZO | 85.86 | **78.59** | 52.70 | 39.01 |
| | | Sign | **86.16** | 76.38 | 53.10 | **39.82** |
| | | Conserve | 85.89 | 77.71 | **53.20** | 39.57 |
| | - | Forward | 84.63 | 76.32 | 53.59 | 40.64 |

### B.4. Memory and Time Efficiency on Larger Transformers

To assess the scalability of `ZeroFlow`, we evaluated its efficiency on two larger vision transformers, ViT-L/16 and ViT-H/14. As shown below, `ZeroFlow` consistently offers substantial memory savings across all model sizes. Notably, even when using ZO-SGD-Sign, the runtime remains faster than that of standard FO optimization.

### B.5. Longer Task Sequence

To further assess robustness, we evaluate performance on an extended task sequence consisting of 20 tasks. As shown below, `ZeroFlow` continue to deliver comparable performance. Additionally, following (Wang et al., 2024), we additionally

Table 7: **Evaluation on lager transformers.**

| Optimizer | ViT-B/16 | | ViT-L/16 | | ViT-H/14 | |
|---|---|---|---|---|---|---|
| | Memory⇓ | Runtime⇓ | Memory⇓ | Runtime⇓ | Memory⇓ | Runtime⇓ |
| FO-SGD | 12.08GB | 59.3s | 33.27GB | 65.0s | 78.09GB | 190.1s |
| ZO-SGD (q=1) | 2.41GB | 32.4s | 3.77GB | 47.0s | 6.45GB | 118.7s |
| ZO-SGD (q=4) | 2.41GB | 111.7s | 3.77GB | 178.3s | 6.45GB | 442.6s |
| ZO-SGD-Sign | 2.41GB | 32.4s | 3.77GB | 48.7s | 6.45GB | 119.3s |
| ZO-SGD-Conserve | 2.41GB | 70.1s | 3.77GB | 108.9s | 6.45GB | 222.3s |
| Forward | 3.94GB | 45.9s | 5.82GB | 142.0s | 9.85GB | 372.5s |

Table 8: **Additional Experimental Results of EASE on 20 Sequential Tasks.**

| Method | Optimizer | Strategy | Avg | Last | FWT | BWT |
|---|---|---|---|---|---|---|
| EASE | SGD | FO | 87.32 | 80.20 | -6.89 | -6.79 |
| | | ZO | 82.65 | 75.98 | -8.33 | -7.71 |
| | | Sign | **83.47** | **76.13** | -8.01 | -7.22 |
| | | Conserve | 82.20 | 75.94 | -8.64 | -7.93 |
| | Adam | FO | 86.67 | 78.19 | -7.17 | -6.80 |
| | | ZO | 84.07 | 76.89 | -7.92 | -7.19 |
| | | Sign | **84.16** | **76.90** | -7.95 | -7.20 |
| | | Conserve | 83.82 | 76.76 | -8.04 | -7.07 |
| | - | Forward | 82.84 | 76.32 | -8.25 | -10.84 |

adopt the **FWT** and **BWT** metrics to assess the overall performance of `ZeroFlow`. FWT (Forward Transfer) quantifies the average influence of prior knowledge on the learning of new tasks, while BWT (Backward Transfer) measures the average influence of learning new tasks on the performance of previously learned $K - 1$ tasks.

$$\text{FWT} = \frac{1}{K-1} \sum_{j=2}^{K} (a_{j,j} - \tilde{a}_j), \quad \text{BWT} = \frac{1}{K-1} \sum_{j=1}^{K-1} (a_{K,j} - a_{j,j}) \tag{10}$$

Here, $a_{k,j}$ denotes the accuracy on task $j$ after training on the $k$-th dataset, and $\tilde{a}_j$ represents the accuracy of a random initialized model trained only on dataset $\mathbb{D}_j$.

