# OpenReview forum: "ZeroFlow: Overcoming Catastrophic Forgetting is Easier than You Think"
_ICML.cc/2025/Conference — ICML 2025 poster_

### Official Review · Reviewer_WA47 · 2025-03-09

**Overall Recommendation:** 3

**Summary:**

This paper investigates continual learning for deep neural network when the gradient is not accessible -- instead of using backpropagation methods (first order methods (FO)), the gradient is approximated by forward pass methods (zeroth-order optimization (ZO)). The paper present ZeroFlow benchmark, where different zeroth-order approaches are proposed (combination of different apprximation method, direction computation and parameter update). The benchmark use ViT-B/16 pre-trained model on IN21K and perform 10 task class incremental learning on multiple datestes (Cifar-100, CUB, ImageNet-A, OmniBenchmark). Based on experimental results, this work proposes 3 different enhancement that can be used to further improve ZO approaches for continual learning.

**Claims And Evidence:**

The claims are clearly supported by the appropriate experimental results. Mostly are easy to follow by the reader, however, some description can be improved for a better readability, e.g. Figure 4 - presenting last-task accuracy, however the bolded description "Results for Forgetting".

**Essential References Not Discussed:**

In Sec. 4.1. when EASE and APER appears, there are no references. One sentence explanation for each method would be also helpful for the reader (or a longer one in different place -- Sec 3?). Currently, the reader just receives the acronyms.

**Experimental Designs Or Analyses:**

Experimental designs follow the standard class-incremental strategy of spiting datasets into multiple tasks. In this case, it's always ten.

**Methods And Evaluation Criteria:**

The proposed methods seem like a comprehensive combination of multiple ZO forward-pass methods. But not an expert here if the selection is good.

For the dataset and evaluation metrics -- this work uses standard CL performance evaluation metrics. However, here, maybe interesting would be to see forward/backward transfer and values for the particular tasks (in appendix). That would be interesting and make the benchmark more informative.

**Other Comments Or Suggestions:**

1. Presenting joint training for all the methods - as an upper-bound.
2. Would be interesting to see if there's a different knowledge transfer between the tasks in ZO vs FO methods. Maybe that can be added to the appendix and one the most interesting plot to the main paper?


Small ones:
1. A mistake in Conclusion in the last sentence, 3 enhancements were proposed, not 2.
2. Formatting of Table 1: what bolding means? Some sections are missing bolding at all - it is confusing for the reader in the current form.

**Other Strengths And Weaknesses:**

Strengths
1. Quite comprehensive benchmark with multiple methods, datasets, and ViT-based backbone.
2. Interesting insights and proposal of further enhancements.

Weaknesses:
1. Not presenting longer sequences than 10 tasks.

**Questions For Authors:**

Enhancements are presented alone, separately posing improvements. It's not clear if they can be combined together and what will be the results. Maybe the authors already tried this and can share the results in the appendix?

**Relation To Broader Scientific Literature:**

The relation to forward-pass only optimization and appropriate methods are well discussed and introduced in Sec.3.

CL literature is well-covered and referenced.

**Theoretical Claims:**

Most of the claims are supported by empirical analysis. No theoretical claims with proofs.

---

> ### Author Rebuttal · Authors · 2025-04-01
>
> **Q1: Caption of Figure 4**
>
> Thanks for pointing out! We've corrected it.
>
> **Q2: Missing Quotes and Descriptions**
>
> We've included quotes (EASE, CVPR-24 and APER, IJCV-24) and use one sentence to describe them.
>
> **Q3: Longer task sequences**
>
> As shown below, we evaluated the results on a task sequence of 20. ZO method still provides comparable performance across learning metrics. Besides, we will add evaluations on three other datasets for further comparison in the appendix.
>
> |  |  |  |  CIFAR-100 (20)  |  | |
> |--------|--------|---|----|---|----|
> |Optimizer|Strategy|Avg|Last|FWT|BWT|
> |**SGD**|FO|87.32|80.20|-6.89|-6.79|
> | |ZO|82.65|75.98|-8.33|-7.71|
> | |Sign|83.47|76.13|-8.01|-7.22|
> | |Conserve|82.20|75.94|-8.64|-7.93|
> |**Adam**|FO|86.67|78.19|-7.17|-6.80|
> | |ZO|84.07|76.89|-7.92|-7.19|
> | |Sign|84.16|76.90|-7.95|-7.20|
> | |Conserve|83.82|76.76|-8.04|-7.07|
> |-|Forward|82.84|76.32|-8.25|-7.84|
>
> **Q4: Knowledge Transfer**
>
> Following [7], we provide the results of BWT and FWT of EASE with 20-split CIFAR-100 datasets above. We'll add these results and analysis into the Appendix of the main paper as an interesting plot. Thanks for your great suggestions to further enhance the quality of our manuscripts!
>
> [7]  A Comprehensive Survey of Continual Learning: Theory, Method and Application, TPAMI 2024.
>
> **Q5: Upper-bound for ZeroFlow**
>
> Thanks for your nice advice. We agree that the upper bound is significantly valuable for ZeroFlow. We added a linear layer for classification after the pretrained ViT-B/16-IN21K backbone and then fine-tune it with several  ZO methods as the upper bound, as shown below. The updated results will be included in Table 1 of the revised manuscript. Moreover, we're opening up a leaderboard to the community, and will offer more upper bound results on various datasets and CL method. Thank you again!
>
> | Joint-training | FO   | ZO   | Sign | Conserve | Forward  |
> |----------------|------|------|------|----------|----------|
> | SGD            | 92.01| 90.66| 91.71| 91.89    | 91.65    |
> | Adam           | 91.97| 91.46| 91.63| 91.50    | 91.65   |
>
> **Q6: Enhancement together**
>
> The results shown below indicate that the proposed enhancements do not conflict with each other and can be integrated. Specifically, FO training benefits from fine-tuning using hybrid ZO optimization, as illustrated in Figure 7. Additionally, it can be further enhanced by utilizing historical gradients and sparsity perturbation to reverse forgetting and achieve stable optimization, as analyzed in R2Q3. Thanks for your great suggestions.
>
> |                | Hybrid | Historical | Sparsity | Avg   | Last   |
> |----------------|--------|------------|----------|-------|--------|
> | FO-SGD         | -      | -          | -        | 61.24 | 51.02  |
> | ZO-SGD         | -      | -          | -        | 57.87 | 48.32  |
> |                | ✔      |            |          | 61.40 | 51.34  |
> |                |        | ✔          |          | 58.90 | 49.04  |
> |                |        |            | ✔        | 59.47 | 49.24  |
> |                | ✔      | ✔          | ✔        | 62.07 | 51.94  |
>
>
> **Q7: Small Typoes**
>
> We've corrected the typo in Conclusion and added explanations for the bold terms in Table 1.

---

### Official Review · Reviewer_Js2L · 2025-03-11

**Overall Recommendation:** 5

**Summary:**

This submission presents a novel benchmark, ZeroFlow, for evaluating overcoming catastrophic forgetting under a gradient ban. The key insight is that forward pass optimization alone can also mitigate forgetting, which challenges the conventional reliance on backpropagation-based optimization. The study evaluates seven forward-pass optimization algorithms across various forgetting scenarios, datasets, and evaluation metrics. The key findings are, (i) Insights into the trade-offs between forgetting and learning; (ii) Effectiveness of forward pass methods for forgetting; (iii) Efficiency and memory management, more practical for CL scenarios; Next, the submission further proposes three new enhancement techniques: hybrid optimization, historical information, and sparsity-induced estimation method, to improve the efficiency and stability of forward-pass.

**Claims And Evidence:**

The claims made in the submission are indeed well-supported by clear and convincing evidence, as follows,
* The visualization of optimization trajectories (Figure 3/6) supports the claim that forward pass methods can effectively balance learning and forgetting.
* The comparable or better performance supports the claim that effectively overcomes forgetting (Table 1/2).
* The proposed enhancements are well-supported by experimental results that demonstrate improved performance in overcoming forgetting (Tables 3/4, and Figure 7).

**Essential References Not Discussed:**

The literature discussed by the authors appears to be sufficiently comprehensive and closely related to the topic. The cited works cover the necessary background and prior findings, providing a thorough context for their contributions.

**Experimental Designs Or Analyses:**

The soundness and validity of the experimental designs and analyses in this submission are well-constructed, such as the comprehensive benchmark experiment (Table 1, Figure 2/4), efficiency analysis (Table 2), and reasonable enhancement designs (Figure 7, Table 3/4). It is commendable that the manuscript provides ample illustrative observations, such as Figures 2/5/6/8/10, which add to its value.

**Methods And Evaluation Criteria:**

The manuscript employs reasonable methods and evaluation criteria, including common datasets in forgetting scenarios (CIFAR-100, CUB, ImageNet-A, and OmniBenchmark), well-recognized metrics (accuracy, last-task accuracy, and forgetting measures, and memory, query, and runtime efficiency).

**Other Comments Or Suggestions:**

The key claim of the manuscript is that forward passes alone are sufficient to mitigate forgetting, which challenges the traditional reliance on BP-based optimization idea. Based on my experience, this work offers a new technological pathway for the CL community. And, the insights provided are quite valuable for understanding this topic.    Overall, I tend to be positive about this work. Given some concerns and weaknesses (Please see Weaknesses and Questions), I'm willing to discuss them in the rebuttal.

**Other Strengths And Weaknesses:**

**Strengths,**
* The manuscript proposes an interesting and practical scenario, that is overcoming catastrophic forgetting under a gradient ban. The concept of the gradient ban is an intriguing idea. In short, this manuscript introduces a new topic to the community that could potentially inspire some upcoming work.
* The authors propose a benchmark to explore the optimization principles of overcoming forgetting under gradient bans. They evaluated a series of methods that enable learning new tasks and retaining old ones using only forward passes. More valuable, the authors provide insightful observations and build their motivation on these findings.
* The authors provide insights into the evolution of gradient trajectories for new and old tasks during the optimization, such as managing task conflicts. These observations are inspiring for understanding how just forward pass balance learning new tasks and retaining old knowledge, and would likely inspire further research in CL.
* The manuscript is logically structured and engagingly written, with clear presentations. Notably, the authors acknowledge the potential risks of this new topic and discuss them thoroughly. Moreover, they also fully explore the advantages of this benchmark, such as reduced memory usage, faster training, and less forgetting.
* Building on their empirical observations, the authors propose three enhancement techniques that achieve promising performance.

**Weaknesses,**
* Compared to the significant contribution of introducing a new technological pathway for the community, the contributions of the three enhancements are somewhat weak but still represent valuable attempts. As I understand it, the mechanisms of Enhancements 1, 2, and 3 seem to also offer shorter training times or less memory usage. I speculate that this efficiency might stem from a reduction in the average queries. Unfortunately, the authors did not conduct such an analysis. I believe completing this would be highly valuable, as this would well-align their observations and the advantages of these optimization principles.
* The different behavior of the optimization principles on new and old tasks are insightful for researchers looking to follow this topic. However, the legends in the figures (Figures 3, 7, and 8) are too small, including the optimization endpoints and the joint optimization center for new and old tasks, this makes it difficult to fully appreciate the insights conveyed by these visualizations.
* The authors clearly explain the motivation for selecting these forward-pass methods. Given my experience, I can easily follow them. However, for those unfamiliar with these optimization algorithms, there may be a learning curve. Providing brief explanations would be beneficial for improving the accessibility of the manuscript.

**Questions For Authors:**

Here are a few questions that interest me,
* Why does Forward-Grad consume slightly more memory than other forward-pass methods?
* Could the authors provide the brief explanations of the optimization principles behind the used forward pass methods in benchmark? (Please see Weakness 3)
* Could the mechanisms of Enhancements 1, 2, and 3 provide shorter total training times or reduced total memory usage? Or do only specific enhancements offer benefits? (Please see Weakness 1)

**Relation To Broader Scientific Literature:**

The submission provides a thorough discussion of the relevant literature. Prior to this study, substantial efforts relied on gradient information to overcome forgetting, they tries a new technical pathway to do this. And their key contribution is overcoming forgetting using the forward pass method, which is related to catastrophic forgetting, and optimization in CL. The authors discuss literature related to these concepts, including CL, FO method, ZO method, and others.

**Theoretical Claims:**

The manuscript includes reasonable theoretical claims, such as the process of gradient estimation (Equation 1, Algorithm 1) and the definition of zeroth-order optimization in CL scenarios (Section 3.2). Moreover, the authors provide empirical evidence to support their theoretical claims (Figure 3).

---

> ### Author Rebuttal · Authors · 2025-04-01
>
> **Q1: More Discussion of Enhancements**
>
> Indeed, as you note, Enhancement 3 has a potential acceleration advantage, which benefits from the reduction in average queries. More analysis see Enhancement 3 in our response 3 to Review HREV. Thank you for your suggestion, which effectively improves our proposed enhancement technique.
>
> **Q2: Visualization of Trajectory**
>
> We fully acknowledge that the small legend hindered the clarity of key insights. We will enhance figure readability by (i) enlarging all legends (old and new task), and (ii) zooming in on optimization endpoints (black, red and blue mark) and joint optimization centers (star mark). These revisions will ensure the visualized insights about task-specific optimization behaviors are more intuitively conveyed.
>
> **Q3: Concise Overview of ZO Series**
>
> Zeroth-order optimization aims to minimize/maximize an objective function $f: \mathbb{R}^n \to \mathbb{R}$ without derivative information. The core problem is formulated as $ \min_{\theta \in \mathbb{R}^n} f(\theta) $, where $\theta$ denotes the optimization variable. To enable gradient-based updates, Simultaneous Perturbation Stochastic Approximation (SPSA) is a commonly used technique to approximate gradients by perturbing the input variables. Specifically, the gradient $ \nabla f(\theta) $ at point $ \theta $ is estimated as:
>
> $\nabla L(\theta, \xi; B) = \frac{f(\theta + \epsilon \xi; B) - f(\theta - \epsilon \xi; B)}{2 \epsilon} \cdot \xi^{-1},$
>
> where $ \xi \sim \mathcal{N}(\mathcal{0}, \mathcal{I}) $ is a random perturbation vector, and $ \epsilon > 0 $ is a small perturbation step size (typically adjusted during optimization).
>
> **ZO-SGD:** Using the gradient estimator $\nabla L(\theta, \xi; B)$, zeroth-order algorithms, such as ZO-SGD, follow the iterative update rule:
>
> $\theta_{t+1} = \theta_t - \alpha_t \nabla L(\theta_t, \xi_t; B),$
>
> where $\alpha_t$ is the learning rate at step $ t $. ZO-SGD bypasses explicit gradient computation through local function evaluations, making it suitable for high-dimensional, non-convex optimization problems.
>
> **ZO-SGD-Sign:** A variant of ZO-SGD, known as ZO-SGD-Sign, improves upon the original approach by approximating the gradient direction using the sign of the gradient estimate. The update rule becomes:
>
> $\theta_{t+1} = \theta_t - \alpha_t \, \text{sign}(\nabla L(\theta_t, \xi_t; B)),$
>
> where $ \text{sign}(\cdot) $ denotes the element-wise sign function. This approach often leads to faster convergence in some problems where the magnitude of the gradient is not as important as its direction.
>
> **ZO-SGD-Conserve:** Another variant is ZO-SGD-Conserve, which conserves the gradient information over multiple iterations. The update rule for this method is:
>
> $\theta_{t+1} = \theta_t - \alpha_t \cdot \frac{1}{k} \sum_{i=0}^{k-1} \nabla L(\theta_{t-i}, \xi_{t-i}; B),$
>
> where $ k $ is the number of past iterations used to average the gradient estimates. This method is beneficial when the gradient updates are noisy, and averaging helps stabilize the optimization process.
>
> **Forward Gradient:** The forward gradient is an approximation technique where the gradient is computed by evaluating the function at a perturbed point and using a linear approximation. Specifically, the gradient $\nabla f(\theta)$ at point $\theta$ is estimated as:
>
> $\nabla f(\theta) \approx \frac{f(\theta + \epsilon \xi) - f(\theta)}{\epsilon} \cdot \xi^{-1},$
>
> where $\xi \sim \mathcal{N}(\mathcal{0}, \mathcal{I})$ is a random perturbation vector, and $\epsilon > 0$ is a small perturbation step size. The forward gradient is generally used when the optimization problem requires accurate directional updates without relying on an explicit derivative.
>
> **Q4: Extra Memory of Forward-Grad**
>
> Forward Gradient requires more memory than ZO-SGD because it involves computing gradients through the Jacobian-vector product, which necessitates storing all intermediate activations during the forward pass. In models like ViT, this includes storing large attention matrices and other intermediate results. In contrast, ZO-SGD only requires two forward passes with perturbed inputs, without needing to store these intermediate activations, resulting in lower memory usage.

---

> > ### Comment · Reviewer_Js2L · 2025-04-03
> >
> > The authors addressed my concerns, given their interesting idea and contribution to understanding the optimization behavior of overcoming forgetting, I am inclined to accept this submission and raise my score.

---

> > > ### Author Response · Authors · 2025-04-07
> > >
> > > Thank you for taking the time to read our rebuttal and for raising your score. We're glad to hear that our response addressed your concerns. We truly appreciate your recognition of our work as interesting and inspiring, and its contribution to the community.
> > >
> > > Gratefully, the experiments suggested by other reviewers have also helped make ZeroFlow more solid. We would be greatly encouraged if our work could benefit researchers working on this topic. Thank you again!

---

### Official Review · Reviewer_HREV · 2025-03-11

**Overall Recommendation:** 1

**Summary:**

Claimed contributions:
 - Contrib 1: benchmark (called ZeroFlow) of continual learning using two previously published strategies: EASE and APER, but only using zero order estimation of descent directions, on vision tasks
 - Contrib 2: insights into the role of forward pass in managing task conflict, and trade-offs between forgetting and memory efficiency
 - Contrib 3: 3 tricks to improve zero order continual learning:
    - Contrib 3A: Hybrid zero order
    - Contrib 3B: Leverage historical gradients
    - Contrib 3C: Sparse update directions: randomly set some directions to zero

**Claims And Evidence:**

Contrib 1 is as far as I can tell a new contribution, but it seems rather limited to only compare 2 strategies: EASE and APER.

I was not able to fully appreciate contrib 2: I assume that it refers to figures 3 and 6 and section 3.2, but the figures are not adequately explained (e.g. what do the axes represent ? what is the setup studied here ?).

I was not able either to appreciate the proposed methods to improve zero order continual learning (contrib 3): the techniques are barely described in section 6 with not enough details to be self contained, whereas their should be discussed in more details with ablation studies in order to benefit the community.

**Essential References Not Discussed:**

Not applicable.

**Experimental Designs Or Analyses:**

From experience, the performance of continual learning techniques at mitigating forgetting is greatly dependent on specific values of hyperparameters. An extensive discussion of the strategy to choose these hyperparameters is currently missing so I don't think the current state of the benchmark provides any reliable conclusion

**Methods And Evaluation Criteria:**

The benchmarked datasets seem reasonable, as they are commonly used in the evaluation of continual learning methods. A limitation is that there are only vision tasks in the benchmark.

**Other Comments Or Suggestions:**

- "gradient ban" is, as far as I can tell, a phrase coined in the paper.
- please define a "forward pass method"
- please define "forgetting measure" in the benchmark
- figure 8: what is the "function value" ?
- "genetic" (section 3) should it be "generic" ?
- discussion regarding the ImageNet input dimension in section 3 which is irrelevant for defining the number of parameters in conv nets (convolution kernels do not depend on the input size)
- in eq. 1, what does it mean to compute the inverse of the vector xi ?
- BP-free and BP-based in section 4.2. This probably refers to backpropagation, but it is never defined, neither used elsewhere in the paper.

Imprecise statements:
 - section 3.2 "ZO optimization for catastrophic forgetting" sounds like forgetting is a desirable property that the method tries to amplify

**Other Strengths And Weaknesses:**

Many statements are rather imprecise, or even do not make sense, and would benefit from proofreading (examples in the next field below). Some definitions are missing and new terms seem to be introduced without much discussion.

**Questions For Authors:**

I would suggest some revisions to make the paper more self-contained and easy to read: limit the number of repeated statements and instead carefully define the material needed to understand the benchmarked methods as well as the proposed improvements.

**Relation To Broader Scientific Literature:**

Previous literature in continual learning and in zero order optimization is adequately mentioned as far as I can tell.

**Theoretical Claims:**

Not applicable: there is no theoretical claim in the paper.

---

> ### Author Rebuttal · Authors · 2025-04-01
>
> **Q1: Extensions to Contrib 1**
>
> We extended the experimental scope to enhance Contrib 1. In detail, we evaluated ZeroFlow on extra strategies: memory replay CL and VLM-CL (see **Q5/7 of Reviewer see4**).
>
> **Q2: Explanation of Contrib 2**
>
> In Section 3.2, ZeroFlow examines how ZO optimization helps mitigate catastrophic forgetting by comparing its optimization landscape to that of FO optimization. Figures 3 and 6 show the optimization paths of both methods as they balance preserving prior knowledge and adapting to new tasks. The axes represent the two most influential feature vectors, with the blue and red "X" markers indicating the optima for old and new tasks, respectively. The black dot represents the learned parameters, initially biased toward the old task, while the black star marks the optimal balance between both tasks. We will explain it in the revision.
>
> Also, **Reviewer Js2L** also provided a clear explanation of Contrib 2 (**Strengths** and **Designs Or Analyses Section**) for your consideration.
>
> **Q3: More Discussion of Enhancements**
>
> **Enhancement 1: Hybrid Optimization** begins by leveraging gradients for fast adaptation to new tasks. Once the parameters are sufficiently close to the optimal solution, ZO is employed to fine-tune the solution, addressing forgetting.} i) FO ensures rapid convergence and efficient adaptation to new tasks, while ZO refines the solution in regions where gradients are unreliable or sparse. ii) The inherent randomness of ZO helps avoid sharp minima, akin to the principles of SAM, fostering more stable and generalizable solutions.
>
> **Enhancement 2: Historical Utilization** employs an online Exponential Moving Average (EMA) to retain the past update information of old task gradients, adjusting them to minimize deviations from the historical direction.} By weighting the historical gradients, it reduces the impact of fluctuations induced by new tasks, effectively alleviating forgetting. Moreover, it enhances the stability of ZO optimization, ensuring smoother convergence and preserving knowledge from old and new tasks.
>
> **Enhancement 3: Sparse Perturbation** introduces sparsity into the ZO by setting a fraction of perturbation dimensions to 0, thereby reducing the number of perturbed parameters.} i) This mitigates the instability inherent in ZO by lowering variance in gradient estimation, leading to more consistent updates. ii) Sparsity reduces overhead, making ZO method more practical for high-dimensional CL settings.
>
> Overall, we will include above into the revision for clarity.
>
> **Q4: Effects of the Hyperparameters from CL**
>
> We provided the results on varying projection dimension $r$ and the trade-off parameter $\alpha$ of EASE. As below, ZeroFlow remain solid, which means that we provided reliable conclusions. We'll provide more analysis in version to ensure reliable conclusions. "‘Cons’ below refers to ZO-Conserve.
>
> | r            | FO_64    | FO_32    | FO_16    | ZO_64    | ZO_32    | ZO_16    | Sign_64  | Sign_32  | Sign_16  | Cons_64  | Cons_32  | Cons_16 |
> |--------------|----------|----------|----------|----------|----------|----------|----------|----------|----------|----------|----------|----------|
> | $\alpha$ 0.3 | 91.11    | 91.02    | 91.37    | 78.25    | 78.14    | 78.97    | 82.77    | 83.16    | 83.84    | 82.77    | 82.15    | 82.19    |
> | $\alpha$ 0.1 | 91.23    | 91.30    | 91.47    | 78.62    | 78.81    | 79.21    | 83.21    | 83.90    | 83.58    | 82.22    | 82.25    | 82.46    |
> | $\alpha$ 0.05| 91.37    | 91.39    | 91.54    | 78.45    | 78.82    | 78.94    | 83.12    | 83.25    | 83.15    | 82.46    | 82.18    | 82.41    |
>
> **Q5: More Precise Statements**
>
> We've carefully reviewed the manuscript, clarifying unclear phrases, as follows,
>
> - Gradient ban: We clarify that we've defined this concept in the original manuscript, see Lines 16, 54 and 40. And we've described its scenario to make the concept easier to understand, see Lines 23 to Lines 40.
> - Forward pass method: this concept refer to gradient-free optimization methods that rely on forward pass, specifically referring to ZO and Forward-Grad method. We'll define it.
> - Forgetting measure: The forgetting measure we used is a common metric in CL [5]. We will define it again for clarity.
> - Function value: It represents the optimization objective in Figure 7, where values approaching zero indicate proximity to the global optimum.
> - $x^{-1}$: denotes the element-wise inversion of the perturbation vector, which is necessary to ensure that the expected value of the gradient estimate aligns with the true gradient when $x$ follows a broader asymmetric distribution [6].
> - Other misc: Small errors are fixed, e.g., the subtitle of section 3.2 corrected to ZO optimization for overcoming forgetting.
>
>  [5] A Comprehensive Survey of Continual Learning: Theory, Method and Application, TPAMI-24.
>
>  [6] Multivariate Stochastic Approximation using a Simultaneous Perturbation Gradient Approximation, TAC 1992.

---

> > ### Comment · Reviewer_HREV · 2025-04-07
> >
> > I don't see the claimed update of the manuscript, did you update the pdf ?
> >
> > Figure 3. is still missing a clear legend, and axes labels, which makes it difficult to parse. Your additional comment in the rebuttal that the axes are the "two most influential feature vectors" raises additional questions: how do you define this influence ? Why is it relevant to observe function space trajectories? There are symmetries in the parameter space: e.g. swap two rows in the weights of a linear layer, and the corresponding 2 columns in the weights of the following linear layer, and you obtain the same function. So how is looking at trajectories in parameter space relevant?
> >
> > Another very vague statement noticed in my last review, at the end of page 4:
> > "[...] ZO methods [...] naturally facilitate the exploration of flat regions in parameter space" => this is never actually checked
> >
> > I am also a bit surprised by the very good reviews given by other reviewers.
> >
> > I dont think in its current state that the paper meets the standards of ICML.

---

> > > ### Author Response · Authors · 2025-04-08
> > >
> > > We would like to kindly remind you that, according to ICML policy, **uploading a revised manuscript during the rebuttal phase is not allowed** (please refer to the ICML Reviewer Instructions for more details). Overall, we have thoroughly revised the paper based on the thoughtful feedback and are committed to **incorporating all the changes in the revised manuscript**, including the broadened experimental scope, detailed expansion on the enhancements, and clarification of vague statements.
> > >
> > > **1. Re-clarify to Fig. 3**
> > >
> > > We would like to respectfully remind you that analyzing optimization trajectories is a common practice in machine learning, particularly continual learning (CL) (***e.g., Fig. 3 in [1]***) and multi-task learning (MTL) (***e.g., Fig. 1 in [2], Fig. 1 in [3], Fig. 2 in [4]***), to study gradient conflicts between tasks.
> > > **Strictly following the setup in [1, 3]**, we visualize the optimization behavior of first-order (FO) and zeroth-order (ZO) methods in overcoming forgetting. Specifically, we consider a two-dimensional parameter space $\theta = (\theta_1, \theta_2) \in \mathbb{R}^2$ with the following individual loss functions:
> > >
> > > **$L_1(\theta) = c_1(\theta) f_1(\theta) + c_2(\theta) g_1(\theta)$ for old tasks**,
> > >
> > > **$L_2(\theta) = c_1(\theta) f_2(\theta) + c_2(\theta) g_2(\theta)$ for new tasks**.
> > >
> > > **Thus, the contour plot in Fig. 3 illustrates the overall objective function, defined as $L = L_1(\theta) + L_2(\theta)$, with the $x$- and $y$-axes representing $\theta_1$ and $\theta_2$, respectively.**
> > > Adherence to [3],
> > >
> > > $f_1(\theta) = \log \left( \max \left( |0.5(-\theta_1 - 7) - \tanh(-\theta_2)|, \; 0.000005 \right) \right) + 6$,
> > >
> > > $f_2(\theta) = \log \left( \max \left( |0.5(-\theta_1 + 3) - \tanh(-\theta_2 + 2)|, \; 0.000005 \right) \right) + 6$,
> > >
> > > $g_1(\theta) = \frac{(-\theta_1 + 7)^2 + 0.1 \cdot (\theta_2 - 8)^2}{10} - 20$,
> > >
> > > $g_2(\theta) = \frac{(-\theta_1 - 7)^2 + 0.1 \cdot (\theta_2 - 8)^2}{10} - 20$,
> > >
> > > $c_1(\theta) = \max \left( \tanh(0.5 \cdot \theta_2), \; 0 \right)$,
> > >
> > > $c_2(\theta) = \max \left( \tanh(-0.5 \cdot \theta_2), \; 0 \right)$.
> > >
> > > Note that the similar visualization of optimizing behavior in overcoming forgetting is shared in [1]. **For clarity, we will add a description of the objective function, legend and axes in all the figures.**
> > >
> > > [1] Embracing Change: Continual Learning in Deep Neural Networks, Cells 2020. (*Citations: 626*)
> > >
> > > [2] Gradient Surgery for Multi-task Learning, NeurIPS 2020. (*Citations: 1208*)
> > >
> > > [3] Conflict-averse Gradient Descent for Multi-task Learning, NeurIPS 2021. (*Citations: 361*)
> > >
> > > [4] Independent Component Alignment for Multi-task Learning, CVPR 2023. (*Citations: 50*)
> > >
> > > **2. Clarification of vague statement to "ZO methods naturally facilitate the exploration of flat regions in parameter space"**
> > >
> > > Zeroth-order gradient estimates are known to be noisy approximations of the gradient (Sec. 4.3 in [5], Sec. 3.3 in [6]). Existing researches (Sec. 3.1 in [7], Sec. 1.1 in [8], Sec. 1 in [9], Sec. 6 in [10]) have demonstrated (both experimentally and theoretically) that injecting noise into the gradient direction can help the algorithm escape bad or spurious local minima. Moreover, **[6] (Sec. 3.3: Zeroth-order updates may help to escape spurious and sharp local minima) explicitly shows that noisy gradients can assist the algorithm in finding flat minima and avoiding sharp local minima.** All these insights suggest that ZO methods **have the potential** to guide models toward better local minima. **We have carefully checked the statements to ensure they are rigorous and well-supported.**
> > >
> > > [5] Randomized Gradient-free Methods in Convex Optimization, Encyclopedia of Optimization 2023.
> > >
> > > [6] Addax: Utilizing Zeroth-Order Gradients to Improve Memory Efficiency and Performance of SGD for Fine-Tuning Language Models, NeurIPS 2024 Workshop.
> > >
> > > [7] Escaping from Saddle Points—online Stochastic Gradient for Tensor Decomposition, COLT 2015.
> > >
> > > [8] How to Escape Saddle Points Efficiently, ICML 2017.
> > >
> > > [9] Toward Understanding the Importance of Noise in Training Neural Networks, ICML 2019.
> > >
> > > [10] Noisy Gradient Descent Converges to Flat Minima for Nonconvex Matrix Factorization, AISTATS 2021.

---

### Official Review · Reviewer_see4 · 2025-03-14

**Overall Recommendation:** 4

**Summary:**

The paper explores the challenge of catastrophic forgetting in continual learning under a gradient ban setting, where gradients information is unavailable. To address this, the authors investigate zero-order optimization methods, which rely only on forward passes without requiring backpropagation. Their key finding is that zero-order optimization can mitigate catastrophic forgetting while improving computational efficiency compared to first-order optimization. The paper proposes three enhancements to further improve zero-order optimization on mitigating catastrophic forgetting: hybrid first- and zero-order optimization, integrating historical gradient to stabilize optimization, random sparsity in gradient estimation.

**Claims And Evidence:**

Yes, the paper’s empirical coverage of ZO variants (sign-based, conservative, etc.) and ablations on query budgets (q=1,2,4…) reinforce the authors main claims claims.

**Essential References Not Discussed:**

No.

**Ethical Review Concerns:**

No ethics concerns.

**Experimental Designs Or Analyses:**

Yes.
For evaluation on EASE and APER models, because the models are based on ViT-B/16 models pre-trained on the ImageNet-21k dataset, it is important to provide justification that the test datasets (CIFAR-100, CUB-200, ImageNet-A, and OmniBenchmark) do not overlap with the pre-training data, or that any overlap has been properly accounted for. This ensures that the reported results are not affected by potential data leakage.

**Methods And Evaluation Criteria:**

Yes. This paper is a benchmark paper, and they authors use metrics including average accuracy, final accuracy, and forgetting.

**Other Comments Or Suggestions:**

none

**Other Strengths And Weaknesses:**

Strengths
- Interesting demonstration that catastrophic forgetting can be addressed without gradient signals, which is valuable for real-world “gradient ban” scenarios.
- Comprehensive coverage of ZO methods (SPSA-based, sign-based, conservative, etc.) and an ablation on query budgets.
- Detailed analyses with standard CL metrics (accuracy, forgetting), plus resource usage.

Weaknesses
- The experimental scope, while broad, focuses mostly on standard classification tasks. Future expansions into more diverse data modalities or multi-domain tasks would strengthen generalizability claims.
- ZO-based training can still be slow if the query budget or the model scale is large (though they partly address this with memory overhead analyses).

**Questions For Authors:**

1. In Table 2, the performance difference between FO and ZO optimization methods on APER is quite marginal, whereas the difference is more pronounced for EASE. Could you provide a possible explanation for this discrepancy?
2. Investigating black-box LLM usage would be an interesting extension, especially given the paper’s references to LLM-as-a-service scenarios. Can the authors comment on this?
3. More examples of how ZO training interacts with memory replay or generative replay strategies might broaden applicability.
4. Potential tests on domain-incremental or cross-modal tasks (e.g., image → text) would help confirm the approach’s general utility.
5. Have the authors tested ZeroFlow on extremely large-scale models (like full ImageNet training or large pretrained transformers), and do the memory/time advantages still hold there?

**Relation To Broader Scientific Literature:**

The paper draws on two lines of work: (1) zeroth-order optimization, especially in black-box or gradient-banned contexts (e.g., black-box LLM APIs, non-differentiable modules), and (2) catastrophic forgetting in continual learning. The authors cite standard CL approaches (e.g., EWC, replay-based methods), plus references on ZO from both classical (SPSA, random gradient-free) and modern contexts (MeZO for large language models). The synergy of these areas is interesting: while gradient-based solutions dominate CL, “ZeroFlow” demonstrates that forward-only methods can provide surprisingly strong performance plus memory savings.

**Theoretical Claims:**

The paper primarily focuses on benchmarking and algorithmic proposals rather than formal proofs. The authors reference known theoretical properties of ZO, such as the expected convergence behavior in high-dimensional spaces, but do not introduce new formal theorems.

---

> ### Author Rebuttal · Authors · 2025-04-01
>
> **Q1: Justification of Datasets**
>
> We follow the typical dataset setup to perform all evaluation [1,2]. In general, any overlap has been properly accounted for and domain gap is further considered (e.g. ImageNet-A and OmniBench are acknowledged to have large domain gap with ImageNet, please refer to [1,2]- Datasets Section). Thanks for your nice suggestion, we've stated it in revision for clarity.
>
> [1] Continual Learning with Pre-trained Models: A Survey, IJCAI-24
>
> [2] Class-Incremental Learning: A Survey, TPAMI-24
>
> **Q2: Concern about the Query Budget**
>
> Actually, ZeroFlow has comprehensively addressed this concern. In details, (i) Consistent Query Budget: ZeroFlow consistently operates under a query budget of 1 across all evaluations, which can be seen in Tables 1/2, Figures 1/2/4 etc. (ii) Enhanced Training Efficiency: The proposed enhancement methods retain the same query budget of 1 (see Figures 7/8, Tables 3/4.) while further accelerating training (see Enhancement 3 in our response 3 to Review HREV). (i) and (ii) strongly support our claim--Overcoming Forgetting is Easier. (iii) As you mentioned, the overhead analyses partly address this concern. Moreover, we provide extra insights on query budgets dynamics in Figure 5 to ensure a full understanding. We expect such an analysis to inspire subsequent work to extend the scope of ZeroFlow. Overall, this concern has been addressed in the original manuscripts.
> Your expertise in raising this issue is much appreciated.
>
> **Q3: Performance Discrepancy**
>
> Although both EASE and APER are prototype-based PTM CL models, EASE incorporates adapter training for each incremental task, wherein the training process builds upon previously trained adapters. In contrast, APER adapts the PTM solely during the initial training stage and subsequently maintains frozen for all following tasks. Consequently, the performance of later tasks in EASE is highly contingent on prior task training, whereas in APER, tasks remain independent of one another. Therefore, if a performance discrepancy arises between ZO and FO optimizers, EASE tends to amplify this gap, whereas APER is only marginally affected.
>
> **Q4: Open Discussion about Black-box LLM**
>
> First, our ZeroFlow naturally extends to LLM-as-a-service scenarios. Our tests on the VLM in the rebuttal (the results see Q6) have demonstrated this potential. Second, the forward-pass method we evaluate could be adapted to optimize prompts or lightweight external adapters through sequential API calls. In short, our findings on memory efficiency and forgetting mitigation potentially address key constraints of LLM deployment. Future work could establish benchmarks for black-box LLM continual learning.
>
> **Q5: Extra Eval. on Memory Replay Method**
>
>  We further offer the performance of ZeroFlow on a typical replay-based method (MEMO [3], replay buffer=2000) to broaden applicability. As shown below, ZeroFlow remains stable in overcoming forgetting. Below, CFR and INA denote CIFAR-100 and ImageNet-A, respectively.
>
> |Optimizer|Strategy|CFR-Avg|CFR-Last|INA-Avg|INA-Last|
> |---------|--------|------|-------|------|-------|
> |SGD|FO|87.43|79.66|53.15|38.97|
> | |ZO|**85.92**|79.00|52.87|35.81|
> | |Sign|85.72|79.10|53.31|38.18|
> | |Conserve|85.86|**79.20**|49.20|36.51|
> |Adam|FO|86.45|76.17|54.06|41.54|
> | |ZO|85.86|**78.59**|52.70|39.01|
> | |Sign|**86.16**|76.38|53.10|39.82|
> | |Conserve|85.89|77.71|53.20|39.57|
> |-|Forward|84.63|76.32|**53.59**|**40.64**|
>
>
> [3] A Model or 603 Exemplars: Towards Memory-Efficient Class-Incremental Learning, ICLR-23 (Spotlight)
>
> **Q6: Potential Tests on VLM-CL**
>
> We evaluated ZeroFlow on continual learning of vision-language model (using MoE4Adapter [4], all training protocols follow [4]). As below, the general utility of ZeroFlow is reconfirmed. Furthermore, we are preparing to open up a leaderboard to the community that will cover more tasks, including PTM-CL, VLM-CL etc.
>
>
> | Method | Strategy | CFR |  |
> |--------------|----------|----------------|----------------|
> | |  | **Avg** | **Last** |
> | | FO | 84.32 | 76.89 |
> | | ZO | 84.27 | **76.91** |
> | MoE4Adapter | Sign | **84.38** | 76.73 |
> | | Conserve | 84.26 | 76.75 |
> | | Forward | 83.96 | 76.53 |
>
> [4] Boosting Continual Learning of Vision-Language Models via Mixture-of-Experts Adapters, CVPR-24.
>
> **Q7: Memory/Time Advantages on Larger Transformers**
>
> Yes, we had employed two larger transformers (ViT-L/16 and ViT-H/14) to evaluate the efficiency of ZeroFlow, as shown below. All of them offer memory advantages, and ZO with ZO-Sign still run faster than FO.
>
> |Opt|Base-Mem|Base-Speed|Large-Mem|Large-Speed|Huge-Mem|Huge-Speed|
> |---|-------|-------|--------|--------|-------|--------|
> |FO|12.08GB|59.3s|33.27GB|65.0s|78.09GB|190.1s|
> |ZO(q=1)|2.41GB|32.4s|3.77GB|47.0s|6.45GB|118.7s|
> |ZO(q=4)|2.41GB|111.7s|3.77GB|178.3s|6.45GB|442.6s|
> |Sign|2.41GB|32.4s|3.77GB|48.7s|6.45GB|119.3s|
> |Conserve|2.41GB|70.1s|3.77GB|108.9s|6.45GB|222.3s|
> |Forward|3.94GB|45.9s|5.82GB|142.0s|9.85GB|372.5s|

---

### Decision · Program_Chairs · 2025-05-01

**Decision:**

Accept (poster)

**Comment:**

This paper received mixed reviews, three accepts (including strong)and one reject. The paper studies continual earning under a gradient ban (where there is no access to gradient information of the model). The paper evaluates a variety of forward methods. The benchmark shows that zero/order optimization can perform continual learning at lower memory requirements. The paper also proposes several techniques to furter improve results based on several insights on ZO methods. Three reviewers appreciate these results, and even consider that this can open new directions for CL research. The negative reviewer found explanations unclear and statements impresice but has low confidence.

The AC has also read the paper, and agrees that explanations can be improved on many occasions. However, the paper presents an insightful evaluations of ZO methods for CL and contains several interesting results. Therefore the AC recommends acceptance of the paper. The authors should incorporate the rebuttal into the final amera ready version.